# CausalAgents: A Robustness Benchmark for Motion Forecasting

## Abstract

As machine learning models become increasingly prevalent in motion forecasting for autonomous vehicles (AVs), it is critical to ensure that model predictions are safe and reliable. However, exhaustively collecting and labeling the data necessary to fully test the long tail of rare and challenging scenarios is difficult and expensive. In this work, we construct a new benchmark for evaluating and improving model robustness by applying perturbations to existing data. Specifically, we conduct an extensive labeling effort to identify causal agents, or agents whose presence influences human drivers' behavior in any format, in the Waymo Open Motion Dataset (WOMD), and we use these labels to perturb the data by deleting non-causal agents from the scene. We evaluate a diverse set of state-of-the-art deep-learning model architectures on our proposed benchmark and find that all models exhibit large shifts under even non-causal perturbation: we observe a 25-38% relative change in minADE as compared to the original. We also investigate techniques to improve model robustness, including increasing the training dataset size and using targeted data augmentations that randomly drop non-causal agents throughout training. Finally, we release the causal agent labels as an additional attribute to WOMD and the robustness benchmarks to aid the community in building more reliable and safe deep-learning models for motion forecasting (see supplementary).

## 1 Introduction

Machine learning models are increasingly prevalent in trajectory prediction and motion planning tasks for autonomous vehicles (AVs) (Casas et al., 2020; Chai et al., 2019; Cui et al., 2019; Ettinger et al., 2021; Varadarajan et al., 2021; Rhinehart et al., 2019; Lee et al., 2017; Hong et al., 2019; Salzmann et al., 2020; Zhao et al., 2019; Mercat et al., 2020; Khandelwal et al., 2020; Liang et al., 2020). To safely deploy such models, they must have reliable, robust predictions across a diverse range of scenarios and they must be insensitive to *spurious features*, or patterns in the data that fail to generalize to new environments. However, collecting and labeling the required data to both evaluate and improve model robustness is often expensive and difficult, in part due to the long tail of rare and difficult scenarios (Makansi et al., 2021).

In this work, we propose *perturbing existing data via agent deletions* to evaluate and improve model robustness to spurious features. To be useful in our setting, the perturbations must preserve the correct labels and not change the ground truth trajectory of the AV. Since generating such perturbations requires high-level scene understanding as well as causal reasoning, we propose using human labelers to identify irrelevant agents. Specifically, we define a *non-causal agent* as an agent whose deletion does not cause the ground truth trajectory of a given target agent to change. We then construct a robustness evaluation dataset that consists of perturbed examples where we remove all non-causal agents from each scene, and we study model behavior under alternate perturbations, such as removing causal agents, removing a subset of non-causal agents, or removing stationary agents.

Using our perturbed datasets, we then conduct an extensive experimental study exploring how factors such as model architecture, dataset size, and data augmentation affect model sensitivity. We also propose novel metrics to quantify model sensitivity, including one that captures per-example absolute changes between predicted and ground truth trajectories and another that directly reflects how the model outputs change under perturbation via IoU (intersection-over-union) without referring to the ground truth trajectory. The second metric helps to address the issue that the ground truth trajectory is one sample from a distribution of many possibly correct trajectories. Additionally, we visualize scenes with large model sensitivity to understand why performance degrades under perturbations.

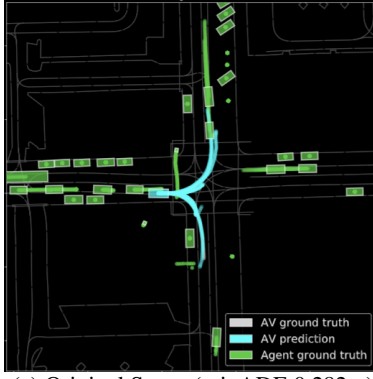
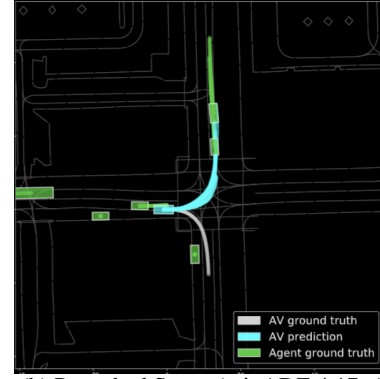

(a) Original Scene (minADE 0.282m)          (b) Perturbed Scene (minADE 4.17m)

Figure 1: **Trajectory prediction is sensitive to removing non-causal agents.** We show a top-down view of a scene from the WOMD (left) and a perturbed version of the scene where we delete all non-causal agents (right). The AV and its predicted trajectories via the Scene Transformer model (Ngiam et al., 2021) are shown in blue, the ground truth trajectory of the AV is grey, and the ground truth of other agents is green. The perturbation causes a large shift in minADE because the model fails to predict the ground truth mode (a right turn), which indicates the brittleness of the model to such perturbations.

Our results show that existing motion forecasting models are sensitive to deleting non-causal agents and can have pathological behavior dependencies on faraway or distant agents. For example, Figure 1 illustrates an original (left) and perturbed (right) scenes with non-causal agents removed. In the perturbed example, the model's prediction misses the right-turn mode, which corresponds to the ground-truth trajectory. Such brittleness could lead to serious consequences in autonomous driving systems if we rely on deep-learning models without further safety assurance from other techniques such as optimization and robotics algorithms. The main contributions of our work are as follows:

1. We contribute a new robustness benchmark for the WOMD for evaluating trajectory prediction models' sensitivity to spurious correlations. We release the causal agent labels from human labelers as additional attributes to WOMD so that researchers can utilize the causal relationships between the agents for robustness evaluation and for other tasks such as agents relevance or ranking (Refaat et al., 2019; Tolstaya et al., 2021).

2. We introduce two metrics to quantify the robustness of motion forecasting models to perturbations, including absolute per-example change in minADE and a trajectory set metric that measures sensitivity without using the ground truth as a reference.

3. We evaluate the robustness of several state-of-the-art motion forecasting models, including Multipath++ (Varadarajan et al., 2021), Wayformer (Nayakanti et al., 2022) , and SceneTransformer (Ngiam et al., 2021). We show that the absolute per-example change in minADE can range from 0.07-0.23 m (a significant $25-38\%$ change relative to the original minADE). We find that all models are sensitive to deleting non-causal agents, and the model with the best overall performance (in terms of regular metrics used to quantify the trajectory prediction performance such as minADE) is not necessarily the most robust.

4. We show that increasing training dataset size and targeted data augmentations that remove non-causal agents can help improve model robustness.

Overall, this is the first work focusing on the robustness of trajectory prediction models to perturbations based on human labels. Such robustness is critical for models deployed in a self-driving car where the reliability and safety requirements are of utmost importance. Ultimately, our goal is to provide a robustness benchmark which can aid the community to better evaluate model reliability, detect possible spurious correlations in deep-learning-based trajectory prediction models, and facilitate the development of more robust models or other mitigation techniques such as optimization and traditional robotic algorithms as complementary solutions to minimize safety risks.

## 2 RELATED WORK

**Robustness evaluation on perturbations.** Machine learning models are known to have brittle predictions under distribution shift. Across multiple domains, researchers have proposed robustness evaluation protocols that move beyond a fixed test set (Recht et al., 2019; Biggio & Roli, 2018; Szegedy et al., 2013; Hendrycks & Dietterich, 2019; Gu et al., 2019; Shankar et al., 2019; Taori

et al., 2020). Evaluation can be broadly categorized into three types: (i) slicing, i.e. existing test data is sliced over multiple dimensions, (ii) perturbations, i.e. existing test data is modified via transformations, or (iii) dataset shift, i.e. new test data is drawn from a different distribution. Our work focuses on perturbations, which have previously been explored in both computer vision and NLP. In computer vision, researchers perturb images via pixel level noise corruptions (Geirhos et al., 2018; Hendrycks & Dietterich, 2019), spatial transformations (Engstrom et al., 2019; Fawzi & Frossard, 2015), and adversarial modifications (Biggio et al., 2013; Szegedy et al., 2013). Such synthetic shifts are easy to apply to arbitrary images, but limited in that they do not test model invariance to more complex modifications such as deleting or modifying irrelevant parts of the image. In trajectory prediction, perturbations are potentially more valuable, since the models train on discrete inputs, namely, the agents and the roadgraph. Because of the structure of the problem, it is easier to reliably construct perturbations that do not modify the ground truth labels. This situation mirrors that of NLP, where sentences composed of discrete words can be modified in ways that do not change the prediction task, and indeed, such transformations have proven valuable for testing the robustness of models and identifying possible biases (Dhole et al., 2021).

**Robustness evaluation for trajectory prediction.** The three types of robustness evaluation (slicing, perturbations, and dataset shift) described above also characterize the trajectory prediction literature. *Slicing.* The most common approach is to slice model performance along different hyperparameters and buckets, such as duration of the historical trajectories (Radwan et al., 2020), size of the training data (Huynh & Alaghband, 2019; Ngiam et al., 2021), sampling frequency (Bera et al., 2016), number of agents in the scene (Rhinehart et al., 2019; Ngiam et al., 2021), criticality / interactivity of the scenarios (Kooij et al., 2019; Ettinger et al., 2021), and speed of the AV (Ngiam et al., 2021). *Perturbations.* Another thread of related work focuses on the robustness of the algorithms to perturbations in *both* training and test data. For example, (Bera et al., 2016) introduced synthetic sensor noises into both the training and test process to evaluate the model's accuracy against sensor noises. (Han et al., 2019) introduced 30% anomalies into the training data (with extra labels), and evaluated the robustness of the algorithm to anomalies in the training process. *Dataset shift.* Less work has focused on dataset shift due to the difficulty of collecting, annotating, and releasing entirely new data. Examples include training and testing in different locations or routes (Schöller et al., 2019; Sun et al., 2021), weather, time of day, and sensor noise (Sun et al., 2021).

A related body of work has studied adversarial robustness for trajectory prediction. In particular, Cao et al. (2022b) propose an adversarial training framework for trajectory predictions as well as domain-specific data augmentations and show that both empirically improve robustness to adversarial attacks. Additionally, researchers have generated more realistic adversarial attack models and benchmarked models against them: Cao et al. (2022a) generates adversarial realistic trajectories using a differentiable dynamic model, Zhang et al. (2022) perturbs existing trajectories to maximize prediction error, Saadatnejad et al. (2022) perturbs trajectories to result in agent collisions, and Wang et al. (2021) simulates directly from sensor data to modify trajectories in a physically plausible manner. Unlike prior work, we evaluate model robustness to "non-causal" domain shifts instead of hard (e.g. weather, location) or adversarial domain shifts. Because these non-causal perturbations are closer to the original validation dataset than the hard domain shifts and do not assume worst-case behavior or presence of an adversary, the discrepancies we observe are in some ways a more immediate priority for improving model robustness.

**Agent relevance.** Since trajectory prediction in AV systems must reason about other agents in the scene, researchers have attempted to efficiently rank agents according to their impact on the AV. The main motivation is to determine which agents to allocate real-time computational resources to. In particular, Aksoy et al. (2020) proposed a driver's saliency prediction model which incorporates an attention mechanism to understand salient features for driving context. Refaat et al. (2019) approximated an agent's influence by looking at the difference between two plans when a given agent is accounted for versus not. However, removing one agent at a time does not account for certain situations where multiple agents may be influencing the car in the same way e.g. two pedestrians are blocking the path of the car and removing one of the pedestrians has no influence on the car. In similar work, Tolstaya et al. (2021) quantifies interactivity using a deep learning model which can suffer from the same robustness issues. More generally, algorithmically defined importance/relevance or interaction scores can be unreliable, especially in scenes with complex interactions between the AV and surrounding agents. In this work, we use human labeling to decide which agents are important,

and our motivation is to use these labels to test model robustness. In the future, our causal agent labels can be used to verify algorithmic definitions of agent importance or relevance.

**Causal reasoning in autonomous driving.** In a similar line of work, Ramanishka et al. (2018) collect causal annotations using human labelers for the Honda Research Institute Driving Dataset and they slice performance of an object detector over scenes with varying causal attributes. Our work instead uses causal labels to evaluate model robustness for trajectory prediction on WOMD, and we provide more fine-grained per-agent measurements of causality.

## 3 METHODS

### 3.1 LABELING CAUSAL AGENTS IN WOMD

The objective of the labeling task is to identify all agents — cars, cyclists, or pedestrians — that are causal to the AV at any time during a driving segment. Although we are more interested in removing non-causal agents from each scene, we ask labelers to identify causal agents since there are typically fewer of them and they tend to be closer to the AV, making them easier for labelers to identify.

**Data.** We focus on labeling the WOMD *validation data* because our primary goal is to evaluate the robustness of models trained on the original dataset. Each example in WOMD is 9.1 seconds in length (91 steps at 10Hz) and is generated in overlapping windows from a 20-second video segment. We label the 20-second segments to give labelers access to a longer time horizon and to not waste resources on labeling overlapping scenes. Moreover, both the regular and interactive WOMD validation sets are generated from the same 20-second segments of data, hence, our causal labels can be used for both.

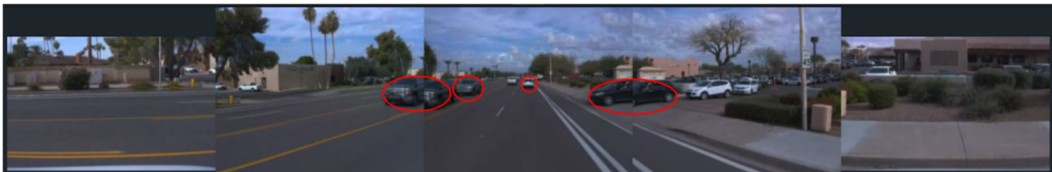

Figure 2: Camera images from a randomly chosen scene in the labeling UI. The causal agents are circled.

**Labeling policy and UI.** Causality is an inherently subjective label since human drivers may vary in their judgements of which agents in the scene affect their decisions. Therefore, we want to be overly conservative and identify as many causal agents as possible to maximize the likelihood that removed agents are actually non-causal. If human labelers are unsure if an agent is causal or not, we instruct them to include it as causal. We emphasize that false positives (identifying an agent as causal when it is truly non-causal) are acceptable to a certain extent, but we should avoid false negatives (failing to identify a truly causal agent). (Appendix A includes the exact instructions given to labelers.) That said, in ambiguous situations, we did not expect labelers to reason about chained causality relationships. For example, if the AV is driving behind a queue of 5 cars and the first car were to brake, it could eventually cause the car in front of the AV to brake. In this situation we would only expect the labeler to identify the car directly in front as causal.

The labeling UI is a web-based 3D view of the AV and its surroundings in the 20-second segmented videos. An example is shown in Fig. 2 where the camera images from a randomly selected scene overlaid with the causal annotations provided by the human labelers.

**Human annotations.** To maximize coverage and avoid false negatives, each scene is annotated by 5 human labelers and we designate causal agents as all agents that *any labeler* identified as causal. Appendix B shows the distribution over causal agents for the number of human labelers who selected the agent as causal. The majority of causal agents are selected by all 5 labelers, but a significant portion (24%) are selected by only 1 labeler.

### 3.2 CAUSAL AGENT STATISTICS

To understand the properties of causal agents, we compute several statistics of causal agents in the WOMD validation dataset, including the percentage of causal agents (Figure 3a), the distribution of the relative distance between the AV and the causal agents versus all surrounding agents (Figure 3b), and the breakdown of causal versus all surrounding agents by agent type (vehicle, pedestrian, or cyclist) (Figure 3c). Figure 3a shows that the majority of agents are actually non-causal: on average, only 13% of the total agents in the scene are labeled as causal, and 93% of scenes have less than 30%

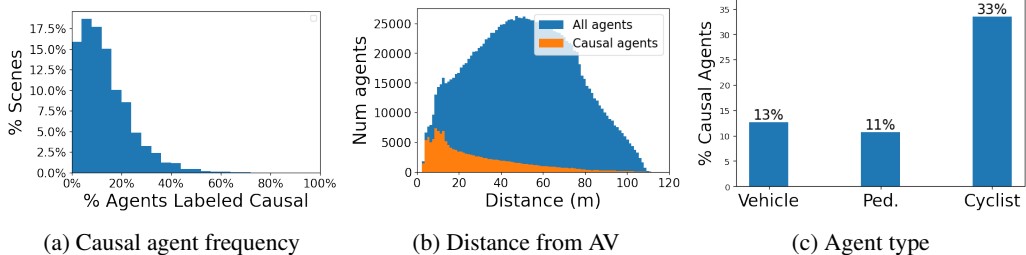

(a) Causal agent frequency

(b) Distance from AV

(c) Agent type

Figure 3: **Causal agent statistics.** Causal agents are less frequent than non-causal agents (on average 13% of agents are causal), and, compared to typical agents, they tend to be closer to the AV. Cyclists are relatively more likely to be causal agents than pedestrians or vehicles.

of the agents labeled as causal. Figure 3b shows that causal agents are typically closer to the AV than non-causal agents; causal agents are an average distance of 28.4m from the AV, compared to an average of 49.4m over all agents. Figure 3c shows the likelihood that an agent of a given type (Vehicle, Ped, or Cyclist) is causal. Surprisingly, cyclists are more likely to be causal agents than any other agents, and vehicles are more likely to be causal than pedestrians. We hypothesize that this is because cyclists usually share the road with the AV and have a strong prior of not respecting road boundaries like a car, whereas there are many parked cars that are not necessarily interacting with the AV and similarly pedestrians can be off the road on sidewalks.

### 3.3 PERTURBED DATASETS

In this work, we consider perturbations that modify the scene by deleting agents. While it is possible to create more complex perturbations, such as adding noise to the xyz position of the agents, we start with deletion since it directly reflects the models' robustness regarding the causal relationships of agents in the scene. Object track states in the WOMD consist of the object's states (e.g., 3D center point, velocity vector, heading), as well as a valid flag to indicate which time steps have valid measurements. To delete an agent from the scene, we set its valid mask to false throughout all time steps (and we double check for each model implementation that all agent state is ignored if the valid bit is false). We consider four different perturbations:

1. **RemoveNoncausal**: Removes all non-causal agents in the dataset.

2. **RemoveNoncausalEqual**: Removes an equal number of randomly selected non-causal agents as there are causal agents in the scene. For example, if a scene has 5 causal agents, we randomly remove 5 non-causal agents. RemoveNoncausalEqual is meant to be a less aggressive form of RemoveNoncausal since it deletes fewer agents and it allows us to compare to RemoveCausal when controlling for the number of agents deleted.

3. **RemoveStatic**: Removes agents whose xyz positions do not change above a certain threshold (e.g. parked cars). We use a threshold of .1 m on the L2 distance of the agent's xyz state to account for sensor noise. Not all static agents are non-causal.

4. **RemoveCausal**: Removes all *causal* agents; the complement of RemoveNoncausal.

Among them, we categorize both RemoveNoncausal and RemoveNoncausalEqual as "non-causal" perturbations. Specifically, to define non-causal perturbations, let us assume $X$ is a scenario representation, $Y$ is the ground truth trajectory of the AV, and $f$ is the ground-truth model that gives the relationship between $X$ and $Y$. If a perturbation $\Delta X$ satisfies $f(X + \Delta X) = f(X) = Y$, we define it as *non-causal perturbation* since it does not impact the relationship between $X$ and $Y$. We define a deep learning model $\hat{f}$ to be robust to non-causal perturbations if $\hat{f}(X + \Delta X) = \hat{f}(X) = \hat{Y} \ \forall$ *non-causal* $\Delta X$, where $\hat{Y}$ is the predicted trajectory from the model. Additionally, we consider RemoveStatic as an important baseline that does not require the human labels. We can thus apply it to the training dataset, which we explore in Section 4.3. Finally, we include the RemoveCausal perturbation as a sanity to ensure models are sensitive to deleting causal agents.

### 3.4 EVALUATION

Since we only have camera and LiDAR data from the AV perspective, we only collect causal labels and evaluate model predictions for the AV trajectory. We report the average minADE (Ettinger et al., 2021), or minimum Average Displacement Error, which computes the L2 norm between the ground truth trajectory and the model's closest prediction, over 3, 5 and 8 seconds on both the original

and perturbed datasets. We measure minADE in units of meters. In all instances, we use the top 6 trajectories for each model (K=6).

**Robustness Metrics.** Since we found in our results that the perturbed minADE often improves for a large fraction of the examples, averaging over examples cancels out some of the effects we would like to measure. Thus, we introduce a metric to measure the per-example absolute change in minADE:

$$\text{Abs}(\Delta) = \frac{1}{n} \sum_{i=1}^{n} |\text{perturbed\_minADE}(i) - \text{original\_minADE}(i)| \tag{1}$$

We report Abs($\Delta$), the standard deviation of Abs($\Delta$), and the the relative percentage change in Abs($\Delta$) with respect to the original minADE. Finally, since the ground truth may represent only one of several correct ways to drive, in Section 4.2 we also consider pairwise differences between the original and perturbed predictions to measure model sensitivity.

| Model | Architecture | Coord. System | # Params |
|---|---|---|---|
| MultiPath++ | LSTM | agent-centric | 125M |
| SceneTransformer | factorized attention transformer | global | 15M |
| Wayformer | early fusion attention transformer | agent-centric | 42M |

Table 1: We evaluate on a diverse set of models.

**The IoU-based metric.** The IoU-based trajectory metric is computed as follows: given two predicted trajectory sets (with and without perturbtion), we first upsample all predicted trajectories (6 of them in each set) to 100Hz, and then voxelize them into a 2D top down grid with resolution of 0.5 meters. We then count the number of voxels both sets occupy, divided by the total number of voxels either output set occupies. To simplify computation, we explicitly ignore the probabilities and speeds of trajectories. This measure quantifies "how geometrically different the trajectories look". An IoU of 1 means the trajectories did not meaningfully change, and an IoU of 0 means the trajectories do not overlap at all. While more complicated versions of this metric could be computed (e.g. earth movers distance), we found this metric intuitive and useful for finding interesting shifts due to perturbation.

**Models.** We select three representative deep learning models for evaluation: MultiPath++ (Varadara-jan et al., 2021), Scene Transformer (Ngiam et al., 2021), and Wayformer (Nayakanti et al., 2022). Importantly, we only consider non-ensembled models (Multipath++ reports ensemble results in their paper and on the WOMD leaderboard). Table 1 reviews the architectural differences and parameter counts of the models. Since we only evaluate on the AV, we typically only train the models to predict the AV, but for MultiPath++ and SceneTransformer we also train models on all agents (which we indicate by appending –All to the model name). Additionally, for SceneTransformer–All, we include both the marginal and joint models (these models are the same when *training* on only the AV.)

## 4 RESULTS

### 4.1 MODEL SENSITIVITY TO NON-CAUSAL PERTURBATIONS

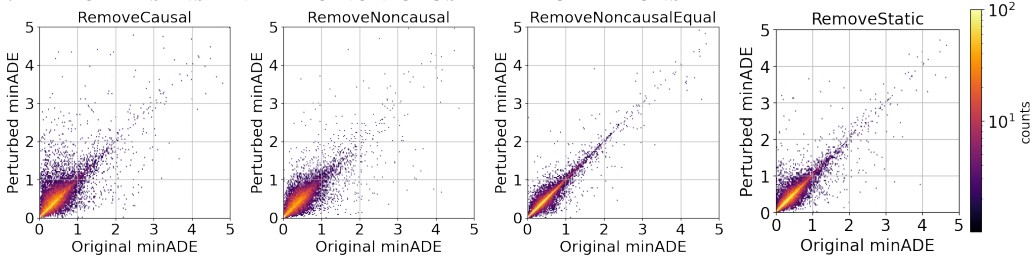

Figure 4: **Model sensitivity to different perturbation types.** We plot the per-example perturbed versus original minADE for all perturbations for the MP++ model. The example frequency is shown with a log color scale where yellow is the most frequent. The majority of examples show minimal change from the perturbation and lie close to the $y=x$ axis. However, across all perturbation types, *there is a long tail of examples that show relatively large change in minADE($>1m$)*. Surprisingly, even for the RemoveCausal perturbation, the *model performance often improves on the perturbed examples*. Comparing RemoveNoncausal and RemoveNoncausalEqual indicates that *the model is more sensitive to removing larger numbers of non-causal agents)*.

In order to understand model sensitivity on a per-example level, Figure 4 plots the perturbed versus original minADE across each perturbation for MultiPath++ (see Appendix H for other models). For

each perturbation type, we observe that the majority of examples show minimal change (i.e. are clustered around the $y=x$ axis), but a long tail of outlier examples experience a large change ($>1m$ in minADE). Among perturbation types, the model is most sensitive to RemoveCausal, which is expected since removing causal agents can change the correct ground-truth trajectory. Interestingly, models are significantly more robust to RemoveNoncausalEqual than RemoveNoncausal, which means removing more agents increases model sensitivity. When comparing RemoveCausal and RemoveNoncausalEqual, which controls for the number of agents removed, we see that the model is significantly more sensitive to removing causal agents than removing non-causal agents.

Surprisingly, across all perturbation types, including RemoveCausal, *the model sees a large portion of examples where minADE improves (i.e. the minADE itself becomes smaller under the perturbation)*: 42.7% of examples show an improvement under the RemoveCausal perturbation, 43.0% for RemoveNoncausal, 49.6% for RemoveNoncausalEqual and 51.0% for RemoveStatic. This finding is counter-intuitive and motivated us to measure model sensitivity in terms of Abs($\Delta$), defined in Equation 1. Across all models, the average Abs($\Delta$) is 0.1450 for RemoveCausal, 0.131 for RemoveNoncausal, 0.051 for RemoveNoncausalEqual, and 0.089 for RemoveStatic. Appendix C reports Abs($\Delta$) for each individual model and perturbation type.

**Comparing models.** Focusing on RemoveNoncausal, in Table 2, we evaluate each model and report the original minADE, perturbed minADE, Abs($\Delta$), the standard deviation of Abs($\Delta$), and $\frac{\text{Abs}(\Delta)}{\text{minADE}_{\text{Ori}}}$ (see Appendix C for other perturbations.) SceneTransformer Marginal shows the lowest average absolute sensitivity, while MultiPath++–All shows the lowest sensitivity *relative* to original minADE. In general, Abs($\Delta$) decreases with the original minADE, but there is no clear relationship between relative Abs($\Delta$) and minADE. Unexpectedly, the marginal SceneTransformer is more robust than the joint (we hypothesize that jointly modeling agents in the scene causes the model to pay more attention to non-causal agents). In Appendix G, we report minFDE, overlap rate, miss rate, and mAP.

| Model comparison, RemoveNoncausal, minADE, $\Delta$ = Perturbed - Original | | | | | |
|---|---|---|---|---|---|
| Model | Original | Perturbed | Abs($\Delta$) | Std. Abs($\Delta$) | $\frac{\text{Abs}(\Delta)}{\text{minADE}_{\text{Ori}}}$ (%) |
| MultiPath++ | 0.376 | 0.395 | 0.141 | $\pm 0.21$ | 37.5% |
| SceneTransformer Marginal | 0.250 | 0.265 | 0.067 | $\pm 0.12$ | 26.8% |
| Wayformer | 0.393 | 0.406 | 0.101 | $\pm 0.16$ | 25.7% |
| MultiPath++-All | 0.900 | 0.945 | 0.226 | $\pm 0.32$ | 25.1% |
| SceneTransformer-All Joint | 0.493 | 0.504 | 0.170 | $\pm 0.26$ | 34.5% |
| SceneTransformer-All Marginal | 0.305 | 0.328 | 0.081 | $\pm 0.14$ | 26.6% |

Table 2: **Model sensitivity to the RemoveNoncausal perturbation**. The SceneTransformer Marginal model shows the lowest average absolute sensitivity to the perturbation, while the MultiPath++–All model shows the lowest sensitivity *relative* to original minADE. Original and Perturbed are the average minADE across the whole dataset. Abs($\Delta$) is the average per-example absolute difference between perturbed and original minADE.

**Slicing the robustness metric.** We further slice the robustness of the models (Abs($\Delta$)) along several dimensions: AV's current speed, the percentage of removed non-causal agents (the number of removed non-causal agents divided by the number of all context agents), and the minimum distance from the AV to removed non-causal agents. The full results are given in Figure 12 in Appendix E. We see that, across all models, model sensitivity increases when we drop a larger fraction of non-causal agents and when the speed of the AV is greater. We also see that model sensitivity typically decreases when we drop agents that are farther away from the AV, though the SceneTransformer models have much noisy robustness measurements when dropping far away agents.

**Visualizing examples.** We also visualize some examples with the largest output changes under the RemoveNoncausal perturbation in Appendix E. The findings are discussed in Section 5.

## 4.2 SENSITIVITY VIA AN IOU-BASED TRAJECTORY SET METRIC

To directly measure the magnitude of model output changes with and without perturbation, we use an IoU (intersection-over-union) based metric (defined in Section 3.4) to compare model sensitivity to different perturbations. The results of three AV-only models under perturbations RemoveCausal, RemoveNoncausal and RemoveNoncausalEqual are shown in Figure 5. We find that models are least sensitive to RemoveNoncausalEqual, and much more sensitive to RemoveCausal and RemoveNoncausal. This is consistent with our finding in Section 4.1, indicating the model is more sensitive to large perturbations since there are more non-causal agents than causal ones in most examples.

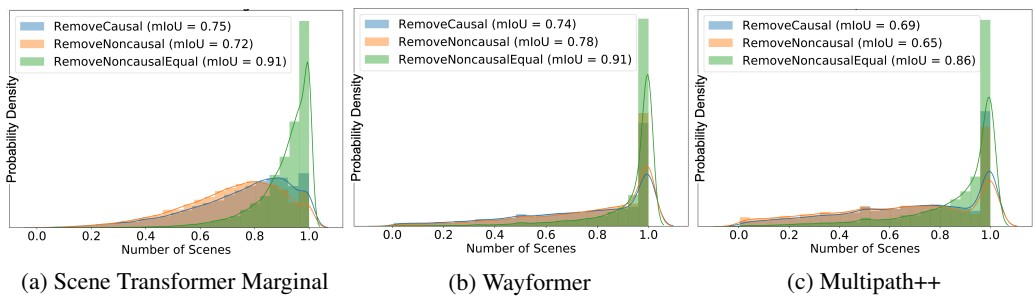

| (a) Scene Transformer Marginal | (b) Wayformer | (c) Multipath++ |

Figure 5: Density distribution of the per-scene trajectory set IoU values for AV-only models under perturbations (RemoveCausal, RemoveNoncausal, and RemoveNoncausalEqual): models are least sensitive to RemoveNoncausalEqual, and more sensitive to RemoveCausal and RemoveNoncausal.

### 4.3 TRAINING WITH DATA AUGMENTATIONS IMPROVES MODEL ROBUSTNESS

We experiment with two types of data augmentation: 1) data augmentations that use a heuristic definition of non-causal agents, such as randomly dropping any static context agent[1], and 2) robustness-targeted data augmentations that directly drop only non-causal agents using a labeled portion of the val set.

**Heuristic data augmentation.** The benefit of using a heuristic definition of non-causal agents for data augmentations is that it can be applied without collecting causal labels. We implement 2 types of heuristic-based data augmentation in the training set of WOMD: Drop Context (randomly dropping context agents) as a baseline, and Drop Static Context (randomly dropping static context agents). We use the MultiPath++–All model and we set the probability of dropping an agent to 0.1 (the best one among 0.1, 0.5, and 0.8). Table 3 summarizes the results for the RemoveNoncausal perturbation (for the per-scene distribution of model sensitivity, see Appendix K). Models with data augmentation show less sensitivity to the perturbations, and, in particular, Drop Static Context shows a significant improvement in minADE and Abs($\Delta$) over Drop Context. We hypothesize that Drop Static Context does better because the static context agents are less likely to be causal. Overall, the results for Drop Static Context imply that dropping non-causal agents via data augmentation in training can improve model robustness to such perturbations at test time.

**Heuristic Data Augs, RemoveNoncausal, minADE**, $\Delta$ = Perturbed - Original

| Model | Original | Perturbed | Abs($\Delta$) | Std. Abs($\Delta$) | $\frac{\text{Abs}(\Delta)}{\text{minADE}_{\text{Ori}}}$ (%) |
|---|---|---|---|---|---|
| MP++-All | 0.900 | 0.945 | 0.226 | $\pm 0.32$ | 25.1% |
| MP++-All Drop Context | 0.948 | 0.988 | 0.209 | $\pm 0.31$ | 22.0% |
| MP++-All Drop Static Context | 0.819 | 0.837 | 0.183 | $\pm 0.26$ | 22.3% |

Table 3: **Heuristic data augmentations.** We compare the MP++-All baseline model to the same model trained with either dropping context agents or dropping static context agents, finding that data augmentations that drop agents that are more likely to be non-causal can improve robustness.

**Non-causal data augmentations.** Motivated by our results that dropping static context agents improves model robustness, we further explore using non-causal perturbations as a data augmentation strategy during training. We randomly sample approximately 70% of the original validation dataset (i.e. 30k scenes), perturb multiple copies of them via the causal labels, and add the perturbed versions into the training dataset. We leave the remaining 30% of the validation set as a holdout for evaluation. We then train a baseline model on the new training dataset as well as a model that randomly drops non-causal agents (when possible) with probability 0.1. In Table 4, we see that similarly dropping non-causal agents helps improve minADE as well as model robustness.

**Noncausal Augs, RemoveNoncausal, minADE**, $\Delta$ = Perturbed - Original

| Model | Original | Perturbed | Abs($\Delta$) | Std. Abs($\Delta$) | $\frac{\text{Abs}(\Delta)}{\text{minADE}_{\text{Ori}}}$ (%) |
|---|---|---|---|---|---|
| MP++ Baseline | 0.395 | 0.408 | 0.150 | $\pm 0.226$ | 38.0% |
| MP++ Drop Non-causal | 0.373 | 0.389 | 0.138 | $\pm 0.194$ | 37.0% |

Table 4: **Noncausal data augmentation.** We fold a portion of the WOMD validation dataset into the original training dataset and apply data augmentations that drop non-causal agents. On held-out validation data, we find significant improvements in model robustness across all three Abs($\Delta$) metrics.

---

[1]Context agents are agents for which no prediction is required in the WOMD leaderboard.

### 4.4 LARGER DATASET SIZE IMPROVES MODEL ROBUSTNESS

We also evaluate model robustness with increasing training data. We randomly select 10%, 20%, 50%, 80% of the training dataset and train separate models on each split. We sample three datasets for each split and average the performance and robustness of each model. Appendix D summarizes the results. As we increase the training data size, the model performance improves (minADE decreases) and both the absolute and relative robustness improves. Interestingly, previously when varying model architectures, we found that the model with the lowest minADE did not always have the best relative robustness. Here, we see a strong trend: for a fixed model architecture, lowering the minADE by increasing the training data results in lower relative sensitivity.

## 5 DISCUSSION

We now discuss a few hypotheses and initial supporting evidence for why models are not robust to the non-causal perturbations.

**Overfitting.** One reason models may fail to generalize to the non-causal perturbations is that they overfit to spurious correlations in the training data (i.e. features that correlate with certain ground truth trajectories but fail to generalize). In our experiments, we observe that models that overfit on the original training dataset (as measured by increasing minADE on the original validation dataset) are *more sensitive* to the non-causal perturbations (see Fig. 11 in Appendix E). Thus, the more the model overfits to spurious features like the number of parked cars in a faraway parking lot, the less well it generalizes to examples where these features are absent. Data augmentation and increasing dataset size may improve robustness by protecting against overfitting.

**Distribution shift.** Models may fail to generalize to perturbations that are significantly different from any data seen during training. In our results, we observe that the more non-causal agents we remove, the less robust models are. Perhaps certain types of scenes with few agents are relatively rare in the training dataset and the model does not generalize well to the distribution shift. By evaluating on the perturbations, we essentially expose the model to rare scenarios not seen in training. One reason that training with data augmentations via dropping (static) context agents or non-causal agents improves robustness could be that it exposes the model to similar scenes during training.

**Over-reliance on agents instead of roadmap.** A third possible reason that models fail to generalize is that they utilize the non-causal agents to infer the drivable areas instead of using the mapping information in the input (we serve high-definition maps and traffic control signals as input features for all models). Our evidence comes from visualizing examples where dropping non-causal agents creates predictions that disobey the roadgraph rules (see Fig. 10 in Appendix E).

**Data-dependent modes.** Finally, many of the state of the art models (e.g. (Ngiam et al., 2021; Varadarajan et al., 2021)) utilize modes (e.g. straight, left, u-turn, etc.) learned from the data distribution, where the input data influences how the model will utilize its K predictions to minimize its loss function. While effective at minimizing minADE-like metrics, these methods provide no coverage guarantees, and can encourage the model to predict multiple speed profiles for the same mode instead of diverse modes. When we triage examples (see Fig. 9 in Appendix E), we find some of the largest failures come from agent deletions that influence which modes the model predicts, demonstrating a weakness of this approach and the metrics they perform well on.

## 6 CONCLUSIONS

We establish a benchmark and metrics for evaluating the robustness of several state-of-the-art models for trajectory prediction for autonomous driving. We find that most state-of-the-art models (with different model architectures and coordination systems) show significant levels of sensitivity to perturbations that remove non-causal agents, with higher sensitivity when removing a greater number of them. While most examples show minimal change in minADE ($\leq 0.1$ m), there is a long tail of examples that can have large changes ($\geq 1$m and sometimes up to 8m). Surprisingly, removing either causal or non-causal agents can cause a significant fraction of examples to improve their minADE. We also find that increasing dataset size and data augmentation can help improve the model robustness. Overall, our results indicate that current machine learning models for trajectory prediction may not be reliable enough on their own, and careful thought needs to be given to how to integrate such models with non-learning components to make a safe system. Finally, we will publish the causal agent labels as complementary attributes to the WOMD to aid future researchers in building more robust models.

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

**For review purposes, we provide a copy of the causal labels annotations in the supplementary materials. The causal agent labels are released as a TFRecord of causal labels protos (see the file 'causal_label.proto' in the supplementary material). The proto maps scenario id to labeler id to a list of agent ids identified by that labeler.

# A    LABELING POLICY

Below is the exact text given to labelers to define causal agents:

> The objective is to identify all agents - cars, cyclists, or pedestrians - that are causal to the AV at any time. A causal agent is one whose presence would modify or influence human driver behavior in any way.
>
> Causality is an inherently subjective label. If you are unsure if an agent is causal or not, please err on the side of including it. In other words, false positives (identifying an agent as causal when it is truly non-causal) are okay, but we should avoid false negatives (failing to identify a truly causal agent).
>
> If the behavior of a human driver would be modified because of a potential action that an agent is likely to take, then that agent should be causal. On the other hand, if the human driver would drive the same regardless of whether the agent is there or not, the agent is non-causal.

The labeling policy also included several examples scenarios with causal agents identified such as Figure 6.

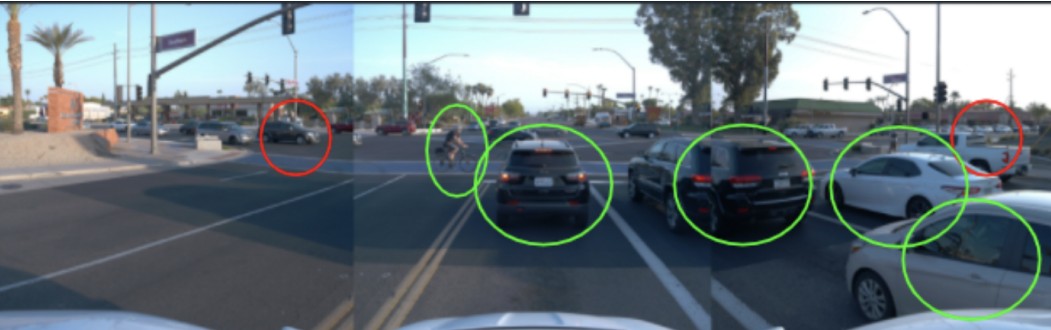

Figure 6: Example from the labeling policy. Causal agents are circled in green and a subset of non-causal agents are circled in red.

# B  LABELER AGREEMENT STATISTICS

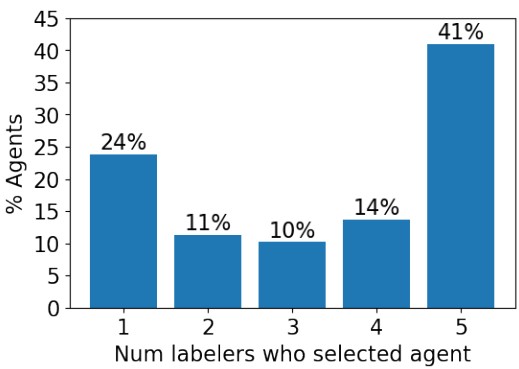

Figure 7: **Labeler agreement statistics.** We plot the distribution of labeler agreement (i.e. number of labelers who selected a given agent) across all agents in the WOMD validation set. The majority of agents are selected by more than one labeler.

# C  MODEL SENSITIVITY TO VARIOUS PERTURBATION TYPES

In this section, we summarize the robustness metrics across different model architectures for each of the perturbation types (RemoveCausal, RemoveNoncausal, RemoveNoncausalEqual, RemoveStatic). We report the model's original minADE, perturbed minADE, average absolute difference between perturbed and original minADE computed per-example (Abs($\Delta$)), standard deviation of Abs($\Delta$), and the relative % change (Abs($\Delta$) divided by the original minADE). Each table below shows the results for a different perturbation dataset.

| RemoveCausal minADE, $\Delta$ = Perturbed - Original | | | | | |
|---|---|---|---|---|---|
| Model | Original | Perturbed | Abs($\Delta$) | Std. Abs($\Delta$) | $\frac{\text{Abs}(\Delta)}{\text{Ori}}$ (%) |
| MP++ | 0.376 | 0.425 | 0.153 | ±0.25 | 40.6% |
| ST Marginal | 0.250 | 0.272 | 0.068 | ±0.14 | 27.1% |
| Wayformer | 0.393 | 0.423 | 0.122 | ±0.20 | 31.1% |
| MP++-All | 0.900 | 0.968 | 0.231 | ±0.34 | 25.7% |
| ST-All Marginal | 0.305 | 0.341 | 0.091 | ±0.16 | 29.7% |
| ST-All Joint | 0.493 | 0.540 | 0.207 | ±0.31 | 42.0% |

Table 5: **Model sensitivity for RemoveCausal, minADE.**

| RemoveNoncausal minADE, $\Delta$ = Perturbed - Original | | | | | |
|---|---|---|---|---|---|
| Model | Original | Perturbed | Abs($\Delta$) | Std. Abs($\Delta$) | $\frac{\text{Abs}(\Delta)}{\text{Ori}}$ (%) |
| MP++ | 0.376 | 0.395 | 0.141 | ±0.21 | 37.4% |
| ST Marginal | 0.250 | 0.265 | 0.067 | ±0.12 | 26.8% |
| Wayformer | 0.393 | 0.406 | 0.101 | ±0.16 | 25.7% |
| MP++-All | 0.900 | 0.945 | 0.226 | ±0.32 | 25.1% |
| ST-All Marginal | 0.305 | 0.328 | 0.081 | ±0.14 | 26.5% |
| ST-All Joint | 0.493 | 0.504 | 0.170 | ±0.26 | 34.5% |

Table 6: **Model sensitivity for RemoveNoncausal, minADE.**

| RemoveNoncausalEqual minADE, $\Delta$ = Perturbed - Original | | | | | |
|---|---|---|---|---|---|
| Model | Original | Perturbed | Abs($\Delta$) | Std. Abs($\Delta$) | $\frac{\text{Abs}(\Delta)}{\text{Ori}}$ (%) |
| MP++ | 0.376 | 0.378 | 0.062 | ±0.12 | 16.4% |
| ST Marginal | 0.250 | 0.252 | 0.023 | ±0.05 | 9.3% |
| Wayformer | 0.393 | 0.395 | 0.042 | ±0.10 | 10.6% |
| MP++-All | 0.900 | 0.907 | 0.103 | ±0.20 | 11.5% |
| ST-All Marginal | 0.305 | 0.308 | 0.025 | ±0.05 | 8.2% |
| ST-All Joint | 0.493 | 0.495 | 0.051 | ±0.11 | 10.3% |

Table 7: **Model sensitivity for RemoveNoncausalEqual, minADE**.

| RemoveStatic minADE, $\Delta$ = Perturbed - Original | | | | | |
|---|---|---|---|---|---|
| Model | Original | Perturbed | Abs($\Delta$) | Std. Abs($\Delta$) | $\frac{\text{Abs}(\Delta)}{\text{Ori}}$ (%) |
| MP++ | 0.376 | 0.387 | 0.094 | ±0.17 | 25.0% |
| ST Marginal | 0.250 | 0.249 | 0.043 | ±0.07 | 17.3% |
| Wayformer | 0.393 | 0.400 | 0.054 | ±0.12 | 13.9% |
| MP++-All | 0.900 | 0.927 | 0.161 | ±0.23 | 17.9% |
| ST-All Marginal | 0.305 | 0.291 | 0.063 | ±0.08 | 20.8% |
| ST-All Joint | 0.493 | 0.462 | 0.118 | ±0.18 | 24.0% |

Table 8: **Model sensitivity for RemoveStatic, minADE**.

To make it easier to compare across perturbation types, we also report the average Abs(Perturbed - Original) minADE for each model and perturbation type in Table 9.

| Abs(Perturbed - Original) minADE | | | | |
|---|---|---|---|---|
| Model | R.Causal | R.Noncausal | R.NoncausalEqual | R.Static |
| MP++ | 0.153 | 0.141 | 0.062 | 0.094 |
| ST Marginal | 0.068 | 0.067 | 0.023 | 0.043 |
| Wayformer | 0.122 | 0.101 | 0.042 | 0.054 |
| MP++-All | 0.231 | 0.226 | 0.103 | 0.161 |
| ST-All Marginal | 0.091 | 0.081 | 0.025 | 0.063 |
| ST-All Joint | 0.207 | 0.170 | 0.051 | 0.118 |
| **Average** | **0.145** | **0.131** | **0.051** | **0.089** |

Table 9: **Abs(Perturbed-Original) across different perturbation types and models**. We report the average absolute difference between the per-example perturbed and original minADE for each model and perturbation type. The model sensitivity for RemoveCausal and RemoveNoncausal is similar, with RemoveCausal resulting in a slightly larger average absolute change. However, when we control for the number of agents and compare RemoveCausal to RemoveNoncausalEqual, we see that the model is significantly more sensitive to removing causal agents than non-causal agents.

# D INCREASING DATASET SIZE

| Train %, minADE, RemoveNoncausal, $\Delta$ = Ptb - Ori | | | | | |
|---|---|---|---|---|---|
| Train(%) | Original | Perturbed | Abs($\Delta$) | Std. Abs($\Delta$) | $\frac{Abs(\Delta)}{Ori}$ (%) |
| 10% | 1.222 | 1.309 | 0.448 | ±0.69 | 37.0% |
| 20% | 1.039 | 1.117 | 0.386 | ±0.53 | 37.2% |
| 50% | 0.947 | 0.996 | 0.266 | ±0.45 | 28.0% |
| 80% | 0.901 | 0.925 | 0.236 | ±0.32 | 26.2% |
| 100% | 0.900 | 0.945 | 0.226 | ±0.32 | 25.1% |

Table 10: **Increasing training data improves robustness.**

# E EXAMPLE VISUALIZATION

## E.1 FAILURE (NON-ROBUST) CASES UNDER NON-CAUSAL PERTURBATION

We have triaged several top sensitive examples under the RemoveNoncausal perturbation. Among these examples, we have found three failure patterns: 1) predictions under the perturbation violate traffic rules, as shown in Figure 8; 2) predictions under the perturbation missed to capture the ground-truth mode, as shown in Figure 9; and 3) predictions under the perturbation violates the causality, for instance, unnecessary slows down when the road becomes more empty due to the removal of non-causal agents, such as the bottom example in Figure 11. Meanwhile, we also have identified examples where the predictions under the non-causal perturbation becomes better, as shown in Figure 10.

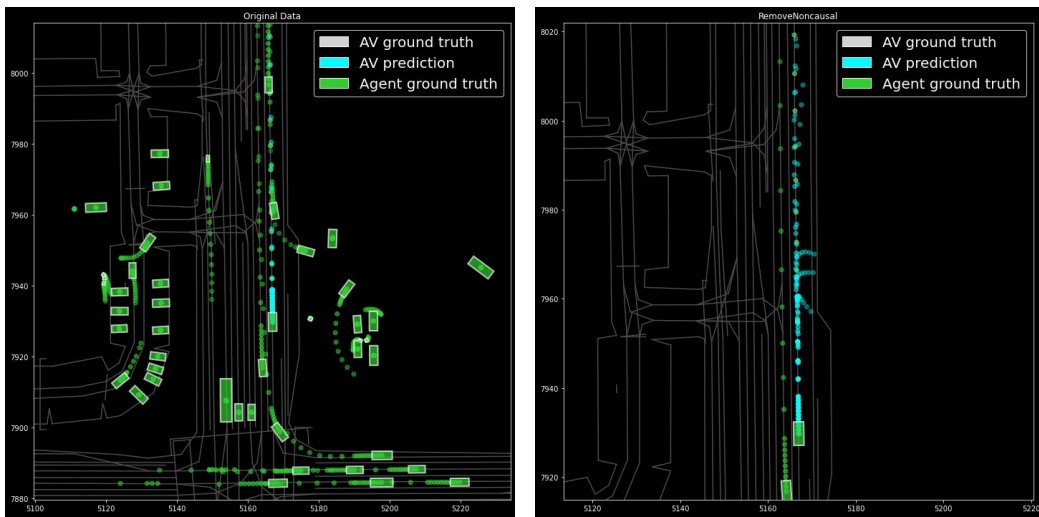

Figure 8: An sensitive example from MP++ under non-causal perturbation: Left side is inference on the original validation data, and right side is inference on the RemoveNoncausal data, where all non-causal agents are removed from the scene, but some of predicted outputs weirdly turn right in the middle of a straight road.

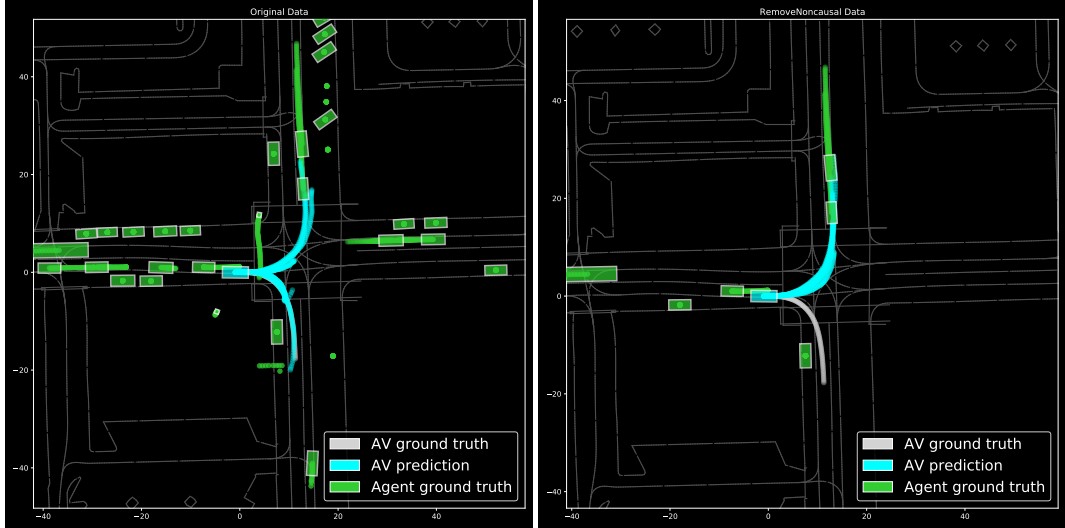

Figure 9: An example from marginal Scene Transformer under non-causal perturbation: Left side is inference on the original validation data, and right side is inference on the RemoveNoncausal data, where all non-causal agents are removed from the scene. It shows a scene where the model performed worse under perturbation, entirely missing the correct mode.

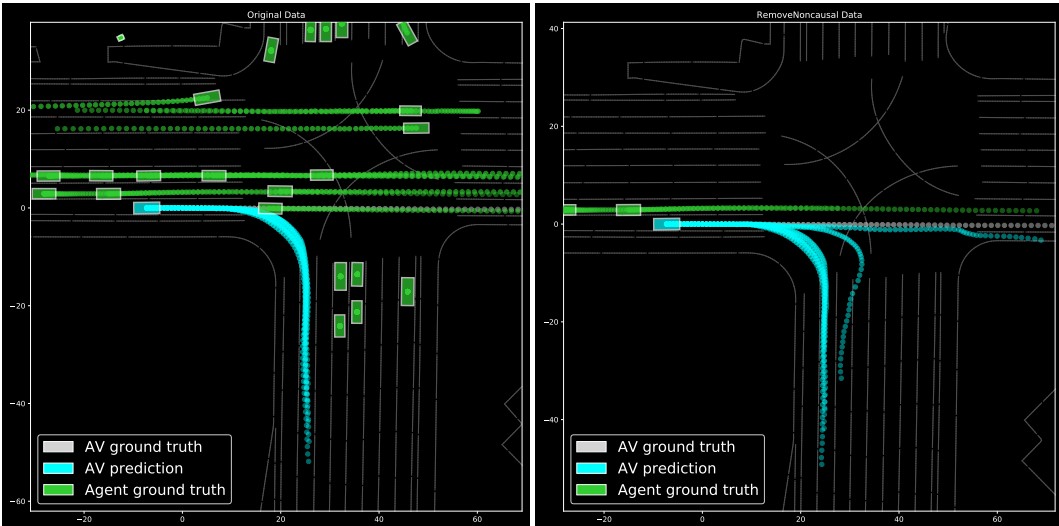

Figure 10: An sensitive example from marginal Scene Transformer under non-causal perturbation: Left side is inference on the original validation data, and right side is inference on the RemoveNoncausal data, where all non-causal agents are removed from the scene. The model outputs under perturbation actually *improved* by capturing the ground-truth mode. The original model missed a mode where it should have driven forward, potentially because of a spurious correlation with the non-causal agent in front of it. After removing that (and other agents), it correctly predicted that mode. However, there is one mode showing a too wide right turn under the perturbation, which is highly unlikely in human driving. This might due to the removal of the static agents from the cross traffic confuses the model about the drivable area.

## E.2 An evidence example for non-robustness due to overfitting

In this section, we show an example scenario that showcases overfitting is one potential reason for poor robustness. We have trained the MP++ with 1M iterations, which overfitted at 210k iterations. We then visualize the predictions of a same example with two different checkpoints, one at 210k iteration and another at 1M. The results are shown in Figure 11. We can see that the robustness of the model under non-causal perturbation becomes bad when the model over-fits.

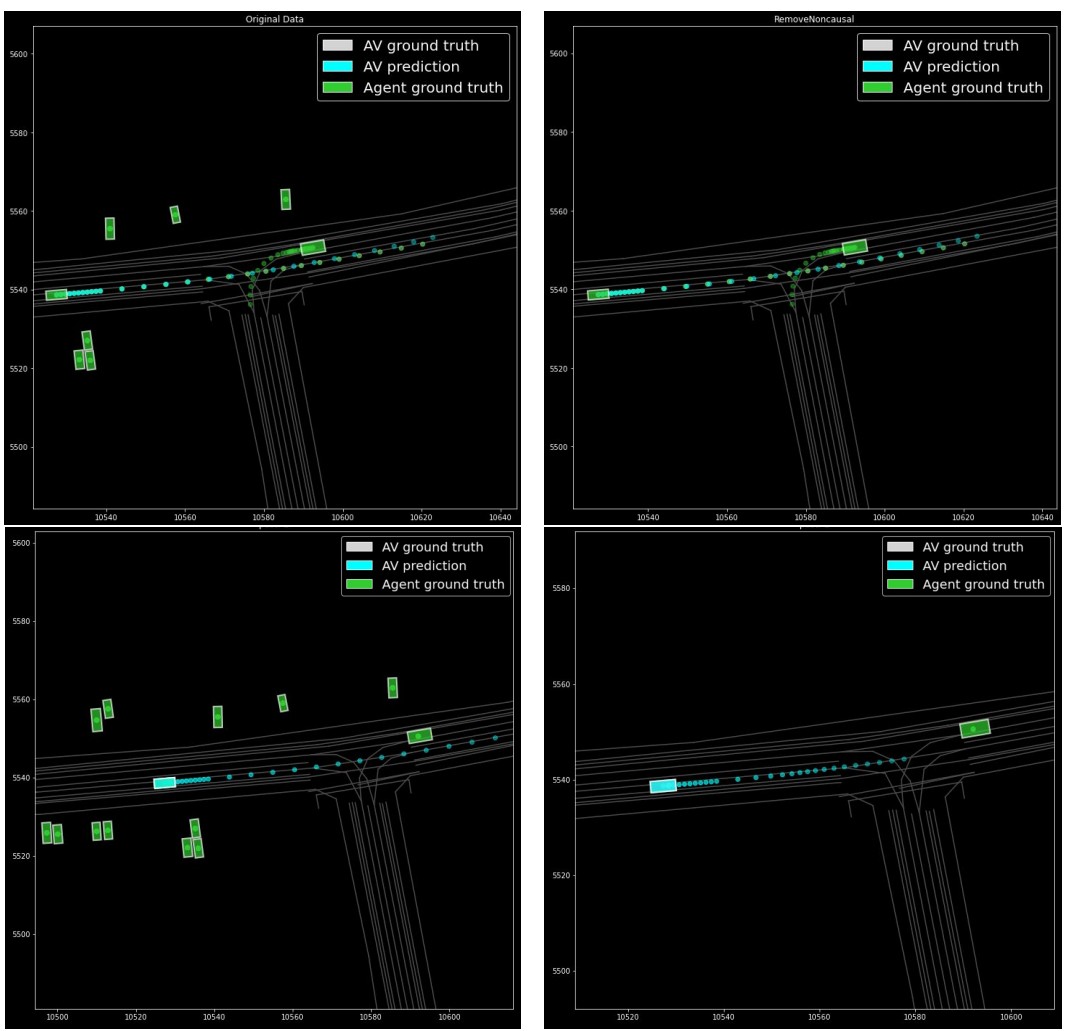

Figure 11: An example from MP++ indicating over-fitting is one possible reason for poor robustness: Left side is inference on the original validation data, and right side is inference on the RemoveNoncausal data, where all non-causal agents are removed from the scene. The top row shows the performance of the model at 210k iteration, while the bottom row is for that of 1M where we observe over-fitting based on minADE on the validation set. We can see that at 1M iteration, the top-1 prediction under the non-causal perturbation unnecessarily slows down. Note that in this plot, we only visualize the top-1 predictions for better visualization. We also only visualize the predictions of the AV in the bottom row.

# F  SLICING RESULTS

We further slice the robustness of the models along several dimensions, including the AV's current speed, the percentage of removed non-causal agents (the number of removed non-noncausal agents divided by the number of all context agents) in the scenarios, and the minimum distance from the AV to removed non-causal agents. Results are given in Figure 12. We found that:

- Along the percentage of removed non-causal agents, we found all models are more sensitive if a larger fraction of agents are removed. Across them, ST Marginal and Wayformer are the most robust models. Compared to Wayformer, ST marginal is less sensitive when more than 40% of the context agents are non-causal and removed from the data (Figure 12 left).

- Along the AV's speed, ST Marginal is more robust when the AV's speed is slower than 45mph (Figure 12 top-left). Note that the high fluctuations at high speed ($> 45$mph) is because we have fewer examples there (the count of examples in each bin is provided in Appendix).

- Along the minimum distance between the removed non-causal agents to the AV, Wayformer is the most robust one, particularly when the minimum distance is larger (i.e., all removed non-causal agent are relatively far away from the AV, Figure 12 right). Such results indicate that Wayformer learns to not pay too much attention to far-away non-causal agents. On the contrary, we noticed that ST models tend to be more sensitive to far-away non-causal agents. We hypothesize that this might be because the global coordination that ST models are used makes it more sensitive to large coordinate values.

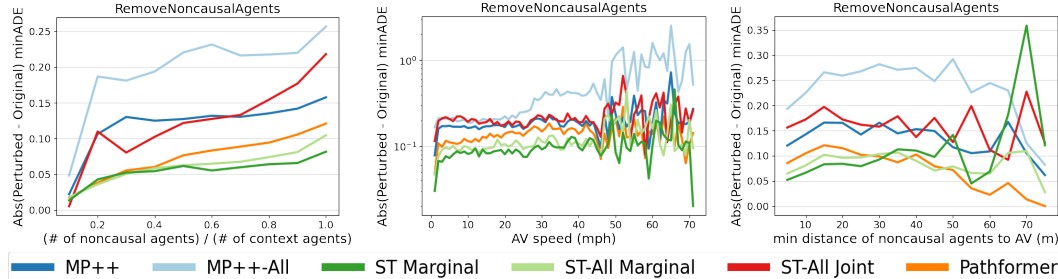

Figure 12: We slice the average Abs(Perturbed - Original) minADE along i) ratio of removed non-causal agents to context agents (left), ii) AV speed (mph) (middle), and iii) minimum distance (m) between the removed non-causal agents and the AV (right).

## G   AGGREGATE RESULTS FOR ALTERNATE METRICS

In this section, we report aggregate results for minFDE, overlap rate, miss rate, and mAP on the RemoveNoncausal perturbation.

| **minFDE**, RemoveNoncausal | | |
|---|---|---|
| Model Name | Original | Perturbed |
| MultiPath++ | 0.853 | 0.895 |
| SceneTransformer Marginal | 0.487 | 0.516 |
| Wayformer | 0.848 | 0.885 |
| MultiPath++-All | 1.430 | 1.551 |
| SceneTransformer-All Joint | 1.170 | 1.176 |
| SceneTransformer-All Marginal | 0.622 | 0.674 |

Table 11: **minFDE RemoveNoncausal results.**

| **Overlap Rate**, RemoveNoncausal | | |
|---|---|---|
| Model Name | Original | Perturbed |
| MultiPath++ | 0.188 | 0.191 |
| SceneTransformer Marginal | 0.211 | 0.210 |
| Wayformer | 0.187 | 0.194 |
| MultiPath++-All | 0.167 | 0.178 |
| SceneTransformer-All Joint | 0.191 | 0.202 |
| SceneTransformer-All Marginal | 0.198 | 0.206 |

Table 12: **Overlap Rate RemoveNoncausal results.**

| **Miss Rate**, RemoveNoncausal | | |
|---|---|---|
| Model Name | Original | Perturbed |
| MultiPath++ | 0.064 | 0.067 |
| SceneTransformer Marginal | 0.059 | 0.064 |
| Wayformer | 0.059 | 0.063 |
| MultiPath++-All | 0.142 | 0.157 |
| SceneTransformer-All Joint | 0.283 | 0.280 |
| SceneTransformer-All Marginal | 0.098 | 0.111 |

Table 13: **Miss Rate RemoveNoncausal results.**

| **mAP**, RemoveNoncausal | | |
|---|---|---|
| Model Name | Original | Perturbed |
| MultiPath++ | 0.554 | 0.524 |
| SceneTransformer Marginal | 0.475 | 0.447 |
| Wayformer | 0.546 | 0.539 |
| MultiPath++-All | 0.268 | 0.243 |
| SceneTransformer-All Joint | 0.227 | 0.223 |
| SceneTransformer-All Marginal | 0.395 | 0.367 |

Table 14: **mAP RemoveNoncausal results.**

# H PER-EXAMPLE SCATTER PLOTS

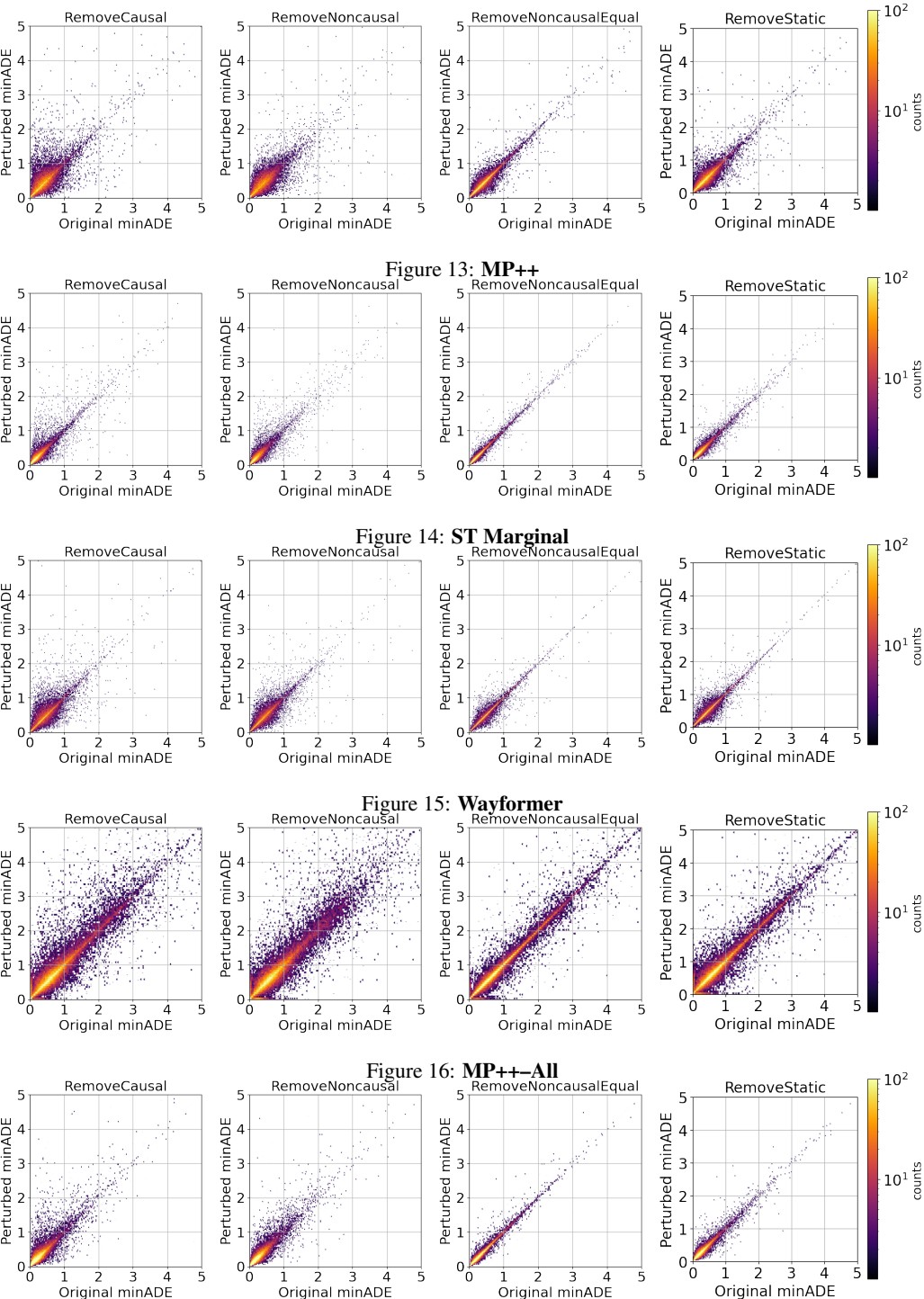

Figure 13: **MP++**

Figure 14: **ST Marginal**

Figure 15: **Wayformer**

Figure 16: **MP++–All**

Figure 17: **ST–All Marginal**

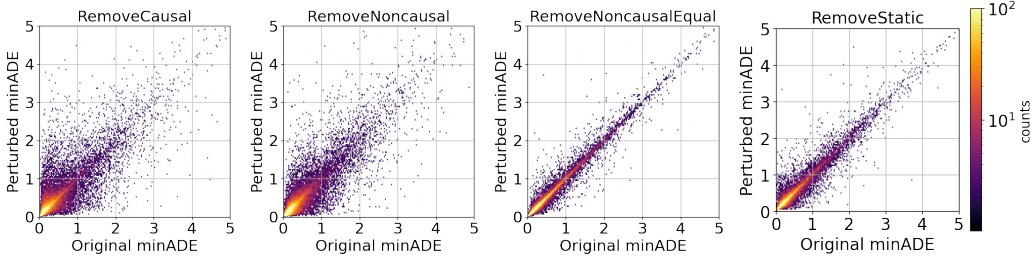

Figure 18: **ST–All Joint**

# I    COMPARISON ACROSS MODELS

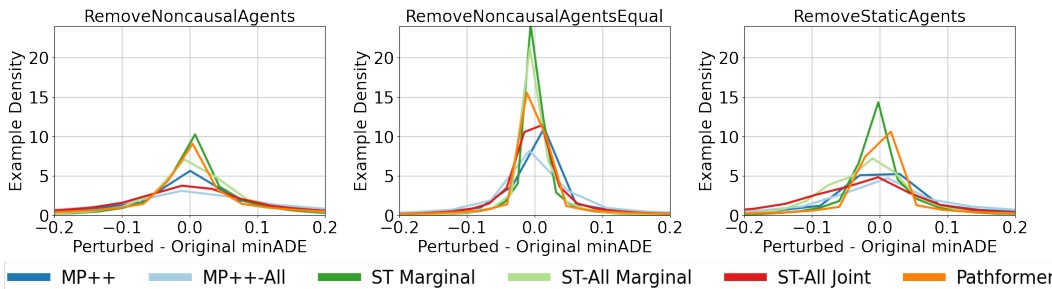

Figure 19: **Models are sensitive to non-causal perturbations.** We plot the distribution of the per-scene difference between perturbed and original minADE for various models and perturbation types. Models that are less sensitive to the perturbation have a higher example density at 0 difference. All models show sensitivity to the non-causal perturbations, which can either increase or decrease the perturbed minADE relative to the original minADE. The models are more sensitive to RemoveNoncausalAgents than RemoveNoncausalAgentsEqual, implying that removing more non-causal agents increases model sensitivity. Among the models, Wayformer and Scene Transformer Marginal show the least sensitivity to the perturbation.

# J    COMPARISON ACROSS PERTURBATION TYPES

Figure 20 shows the sensitivity of the Wayformer AV Only, ST Marginal AV Only, and MultiPath++ All Agents models to each of the perturbation types. The perturbation can either increase or decrease the minADE. On average, it increase the minADE but this depends on the pertubation type (Remove-CausalAgents causes the strongest increase; see Appendix C). The models are most sensitive to both the RemoveCausalAgents and RemoveNoncausalAgents perturbations.

The RemoveCausalAgents has the largest effect on the model, producing the most outliers that increase the difference between the perturbed and original minADE, followed closely by RemoveNoncausalA-gents, then RemoveStaticAgents, and then RemoveNoncausalAgentsEqual. Surprisingly, the sensi-tivity of RemoveNoncausalAgents is close to that of RemoveCausalAgents (exact numbers TODO). However, when we change the number of non-causal agents removed (in RemoveNoncausalAgentsE-qual) to be the same as the number of causal agents removed (in RemoveCausalAgents), the sensitivity is much less.

# K    DATA AUGMENTATIONS

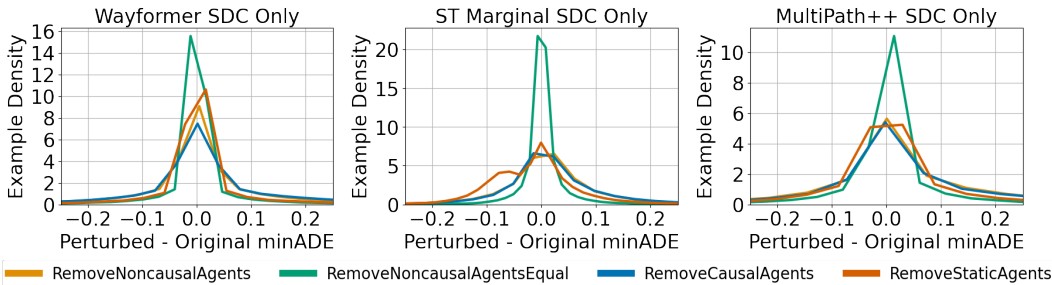

Figure 20: **Models are sensitive to non-causal perturbations**. For three models, we plot the distribution of the per-scene difference between perturbed and original minADE for various perturbation types. The models are least sensitive the RemoveNoncausalAgentsEqual perturbation, and most sensitive (almost equally so) to RemoveCausalAgents and RemoveNoncausalAgents.

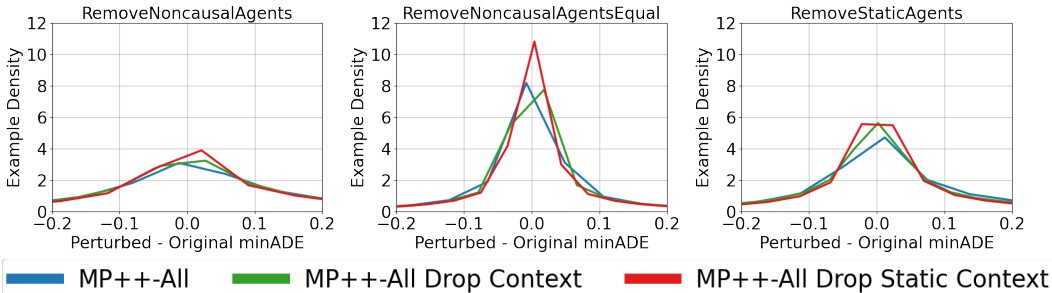

Figure 21: **Targeted data augmentations can improve model robustness.** Dropping static context agents as opposed to context agents has a greater effect on reducing model sensitivity to non-causal perturbations.

## L    TRAJECTORY SET METRICS (CONTINUED)

Since $\Delta$(Ptb-Ori) on minADE only quantifies how the perturbations impact the models' robustness in terms of the distance between ground-truth and the closest predicted trajectories, it does not directly reflect the difference between the two predicted trajectory sets (w/ and w/o perturbations). We thus introduce two *trajectory set metrics* to capture such difference: an IoU based metric as given in Section 3.4 in the main context and a trajectory set minADE defined below. Ideally, a model's predicted trajectory sets would not be sensitive to dropping non-causal agents, meaning we expect a low difference on the trajectory set metrics.

**minADE between trajectory sets (TS_minADE)**. Let $\hat{p}^i_{pert,orig}$ represent the $i$-th predicted trajectory in the predicted trajectory sets w/ and w/o perturbation, respectively. We define TS_minADE = $\min L_2(\hat{p}^i_{orig}, \hat{p}^j_{pert})$, $i, j = 1, 2, \cdots, N$ where $N$ is the number of the predicted trajectories of the model. Hence, a smaller TS_minADE means that two predicted trajectory sets are more similar.

The results for all the models are given in Table 15. We can see that most of the models are sensitive to the RemoveNoncausal perturbation. The Wayformer is least sensitive, which is good. However, it is also least sensitive to RemoveCausal, which indicate that the model is less sensitive to agent removal in general.

| Trajectory set minADE for RemoveNoncausal and RemoveCausal | | |
|---|---|---|
| Model | RemoveNoncausal | RemoveCausal |
| MultiPath++ | 0.037 | 0.024 |
| SceneTransformer Marginal | 0.103 | 0.094 |
| SceneTransformer Marginal-All | 0.140 | 0.141 |
| SceneTransformer Joint | 0.156 | 0.195 |
| Wayformer | 0.0140 | 0.0164 |
| MP++ All Agents | 0.114 | 0.114 |

Table 15: The trajectory set minADE for models evaluated on the perturbations of RemoveNoncausal and RemoveCausal

