# OpenReview forum: "CausalAgents: A Robustness Benchmark for Motion Forecasting Using Causal Relationships"
_ICLR.cc/2023/Conference — Submitted to ICLR 2023_

### Official Review · Reviewer_Chzn · 2022-10-15

**Confidence:** 5
**Correctness:** 2
**Technical Novelty And Significance:** 2
**Empirical Novelty And Significance:** 2
**Recommendation:** 5

**Clarity, Quality, Novelty And Reproducibility:**

- The presentation is clear.

- The data will be released to the public.

**Strength And Weaknesses:**

--- Strengths

- Robustness is essential to the trajectory prediction task. This paper presents many interesting results about how state-of-the-art prediction models perform under perturbations.

- The causal labels that the authors released to the WOMD validation set will be valuable.


--- Weaknesses

- This is an interesting paper. However, my main concern about this paper is that the studies presented in this paper are based on a prediction model whose performance might not be representative. This paper mainly used the MultiPath++-All model for its studies. However, in Table 2, the performance of the MultiPath++-All model is clearly an outlier. It has more than double the prediction error compared to the other models in the table, and it's even a lot worse than MultiPath++. If I understand correctly, the only difference between MultiPath++-All and MultiPath++ is that MultiPath++-All was trained on all agents, so I will expect the performance of MultiPath++-All should be close to or better than MultiPath++ since it's trained on more agents. I hope the authors provide a convincible explanation about why the *-All models have worse performance. The authors used the outlier MultiPath++-All model for most (if not all) of the other studies, including data augmentation and dataset size analysis. I am concerned that the conclusions might not apply to the other state-of-the-art models since the performance of MulthPath++-All seems to be an outlier.

- There are many interesting analyses presented in the paper, but there is not much novelty in the proposed mitigation method. The main method presented in the paper to improve the model's robustness is to augment the training data by randomly removing some static agents. It is not surprising at all that removing some agents will make the model more robust to the removal of agents at test time.

**Summary Of The Paper:**

This paper presents a benchmark and a set of studies about the sensitivity of trajectory prediction models to the removal of non-causal agents.

The authors had human labelers label the agents that influence the ego behavior in the WOMD validation set. They then used those labels to perturb the dataset. They considered four perturbations, removing all non-causal agents, removing some non-causal agents that equaling to the number of causal agents, removing all causal agents, and removing all static agents.

They evaluated three representative trajectory prediction models (MultiPath++, SceneTransformer, and Wayformer) under those perturbations. The results show that all those models are sensitive to the removal of causal agents. For example, the relative mADE delta is around 25%, and the prediction errors even decrease in more than 40% of the samples.

The authors hypothesized that a lack of training data and overfitting could be the cause of this sensitivity. They also found data augmentation (by randomly removing some static agents) could improve the robustness of the models.

**Summary Of The Review:**

This is an interesting paper. However, my main concern about this paper is that the studies presented in this paper are based on a prediction model whose performance might not be representative. And there is not much novelty in the proposed method.

---

> ### Author Response · Authors · 2022-11-11
> **Response**
>
> We thank the reviewer for their feedback and respond to their concerns below.
>
> **Performance of the MultiPath++-All model**. It is easy to explain why the “-All” models have worse performance in general.  We only evaluate model predictions on the AV since we only have causal labels defined with respect to the AV.  Models with the “-All” suffix are trained to predict trajectories for both the AV and other agents in the scene whereas models without this suffix are just trained on the AV.  The models that are just trained on the AV are better at predicting for the AV specifically, and thus do better on our metrics.
>
> While the performance of the MultiPath++-All model is worse, we have no reason to believe it is an outlier or would behave differently than other models.  For example, in Figure 12, we slice the performance of the models over several metrics, and we find that the overall behavior of MultiPath++-All does not vary significantly from the other models (aside from the performance just being worse). We also plot the per-example scatterplots for every model across different perturbation types in Appendix H and find that MultiPath++-All behaves similarly to other models as we vary the perturbation.
>
> **Novelty in the proposed method**. For methods that improve robustness, our contribution is to provide baseline and experimental evidence to the research community that shows that techniques like targeted data augmentations and increased dataset size work in this setting. We also introduce a benchmark that will allow future work to appropriately test new robustness interventions.

---

> > ### Comment · Reviewer_Chzn · 2022-11-12
> > **Thank you for the response. I am still convinced.**
> >
> > > Performance of the MultiPath++-All model
> >
> > MultPath++'s minADE is only 0.376 m, but MultiPath++-All has a minADE of 0.900 m. I do not think training on the additional non-AV agents should increase the prediction errors on the AV agent that much. The AV's behavior should not be that different from the other agents. Including the additional non-AV agents should actually improve the prediction performance of all agents (including the AV agent), if the training is performed correctly. For example, you should make sure you transform the coordinate frame around each target agent when you train on that agent. Without a convincing explanation of the performance of the MultiPath++-All model, I cannot recommend this paper for acceptance since all the following experiments are about improving the MultiPath++-All model performance.
> >
> > I recommend the authors re-do the experiments using the MultiPath++ model.

---

### Official Review · Reviewer_vJxd · 2022-10-17

**Confidence:** 5
**Correctness:** 3
**Technical Novelty And Significance:** 2
**Empirical Novelty And Significance:** 4
**Recommendation:** 3

**Clarity, Quality, Novelty And Reproducibility:**

### **Quality**
The quality of the experiment is high. I have some concerns about the motivation of evaluating robustness with causality as I mentioned in the weakness part. Some important literature is also missing.

### **Clarity**
The presentation of this paper is very clear.

### **Novelty**
The benchmark proposed in this paper is novel.

### **Reproducibility**
The label of the non-causal agent is provided in the supplementary. I believe the results are reproducible.

**Strength And Weaknesses:**

### **Strength**

* Evaluating the robustness of trajectory prediction algorithms is an interesting topic. A standard benchmark is also urgently demanded.

* The structure of the paper is good and the writing is easy to follow.


### **Weaknesses**

* **Correctness of non-causal label.** The correctness of the non-causal label cannot be verified or guaranteed. In particular, the causality of sequential data is usually more complex than expected. For example, Figure 1 does not look correct to me. I find that some important agents (a pedestrian and two vehicles in the opposite direction) are removed. I think calling them “non-causal agents” is not reasonable. Actually, I think it is really hard to identify the non-causal agents unless the underlying causality behind the traffic scenario is explicitly shown. Or interventions are allowed to obtain Randomized Controlled Trials.

* **Missing literature.** Related works about adversarial robustness of trajectory prediction are missing: [1][2][3][4]. Although this paper does not focus on adversarial robustness, I feel that the robustness mentioned in this paper can also be improved by adversarial training.

* **Motivation.** Based on the last point, I question the motivation for evaluating robustness with the non-causal agents. I doubt that the performance drop of prediction algorithms may be caused by the bias and low diversity of the dataset. In contrast, I think adversarial trajectory perturbation may be more fundamental for robustness.

---

[1] Cao, Yulong, Danfei Xu, Xinshuo Weng, Zhuoqing Mao, Anima Anandkumar, Chaowei Xiao, and Marco Pavone. "Robust Trajectory Prediction against Adversarial Attacks." arXiv preprint arXiv:2208.00094 (2022).

[2] Cao, Yulong, Chaowei Xiao, Anima Anandkumar, Danfei Xu, and Marco Pavone. "AdvDO: Realistic Adversarial Attacks for Trajectory Prediction." arXiv preprint arXiv:2209.08744 (2022).

[3] Zhang, Qingzhao, Shengtuo Hu, Jiachen Sun, Qi Alfred Chen, and Z. Morley Mao. "On adversarial robustness of trajectory prediction for autonomous vehicles." In Proceedings of the IEEE/CVF Conference on Computer Vision and Pattern Recognition, pp. 15159-15168. 2022.

[4] Wang, Jingkang, Ava Pun, James Tu, Sivabalan Manivasagam, Abbas Sadat, Sergio Casas, Mengye Ren, and Raquel Urtasun. "Advsim: Generating safety-critical scenarios for self-driving vehicles." In Proceedings of the IEEE/CVF Conference on Computer Vision and Pattern Recognition, pp. 9909-9918. 2021.


**Summary Of The Paper:**

This paper designs a benchmark based on the Waymo Open Motion Dataset for evaluating the robustness of trajectory prediction algorithms. They construct perturbed examples by removing non-causal agents, removing causal agents, removing a subset of non-causal agents, or removing stationary agents. Their experiment results show several state-of-the-art prediction algorithms have performance drops when evaluated on their dataset. They also propose two methods, i.e., increasing the training dataset size and using targeted data augmentation, to increase the robustness.

**Summary Of The Review:**

My biggest concern about the proposed benchmark is the correctness of the causal label. If the correctness of the underlying causality cannot be guaranteed, the evaluation results will be meaningless. In addition, compared to using non-causal agents to evaluate robustness, I think adversarial perturbation is more useful and applicable in real-world AV systems. Overall, I suggest rejecting this paper.

---

> ### Author Response · Authors · 2022-11-11
> **Response**
>
> We thank the reviewer for their feedback and respond to their concerns below.
>
> **Correctness of non-causal label**. We acknowledge that we don’t use an explicit causal model to determine causal labels, and that the labeling process won't be perfect, since the definition of causal agents is subjective (for instance, in Figure 1, the annotators’ opinion differed from the reviewer’s).  However, we use the term “non-causal” because we expect the human annotators to implicitly reason about causality and use the same classification as they would if they were human drivers, which is ultimately what matters for influencing the human driver’s ground truth trajectory.
>
> Moreover, there are clear situations where we don’t need a causal model to identify non-causal agents. For example, cars in parking lots with no access to the main road are always non-causal.
>
> We argue that our labels don’t necessarily need to be provably correct for our testing to find systemic modeling errors. If removing these agents changes the predictions in any way (good or bad), we have a problem. Indeed we have such examples, for instance, Figure 11 in the appendix where removing the parked agents outside of the road slowed down the predictions.
>
> **Missing literature**. We thank the reviewer for pointing out the related work around adversarial robustness of trajectory prediction models. We updated our paper to include a discussion of this work. That said, the focus of our work is on robustness to the least disruptive perturbations, rather than the worst-case of an adversarial modification.
>
> The question of whether techniques that improve adversarial robustness will improve robustness to non-causal perturbations is an interesting one. In image classification at least, robustness interventions designed to improve on adversarial examples do not always transfer to other types of robustness [Taori et. al. Measuring Robustness to Natural Distribution Shifts in Image Classification]. But that may not be the case for trajectory prediction. Regardless, the release of our benchmark and baseline experiments will make it possible to conduct this style of research in the future.
>
> **Motivation**. Demonstrating that our models are not robust to non-causal perturbations highlights an over-reliance on such agents for accurate trajectory prediction. For example, through manual triage, we saw examples where the models relied on the presence of non-causal agents to determine the driveable areas of the road. In low traffic situations where these agents are missing, the model would make less accurate predictions. The non-causal perturbations are arguably more fundamental to robustness since they don’t assume the presence of an adversary or a worst-case scenario for the model. They are a simpler, benignly-generated perturbation that highlight an undesirable brittleness of current models. Moreover, it helps to quantify the possible bad “consequences” caused by the upstream modules’ errors (for instance, if the perception module in an autonomous system missed the detection of some agents) to the predictive models, so that we can mitigate accordingly to assure the safety of the overall system.
>
> With regard to dataset size and bias, one of the findings of our paper is that improving the training set size and using data augmentations increases the model’s robustness to non-causal perturbations. This finding is supported by work on other areas such as image classification; for example, a leading theory for why some recently published models like CLIP have high robustness is that they were trained on larger and more diverse data (Fang et. al. Data Determines Distributional Robustness in Contrastive Language Image Pre-training, ICML 2022).

---

### Official Review · Reviewer_UcfZ · 2022-10-23

**Confidence:** 4
**Clarity, Quality, Novelty And Reproducibility:** 1. The writing is clear and the paper…
**Correctness:** 3
**Technical Novelty And Significance:** 2
**Empirical Novelty And Significance:** 4
**Recommendation:** 6

**Strength And Weaknesses:**

### STRENGTHS:

1. The paper is largely well-written.
2. The empirical analysis is extensive and the paper discusses relevant aspects of model performance.

### WEAKNESSES:

The main weaknesses I find are in dealing with the nature of the problem and the annotation methodology, which I consider are unfortunately crucial towards this paper's contributions.

#### 1\. On Causality and Perturbation:

The community has had a larger discussion on the relationship between perturbations and causality. In explainable AI, perturbations are used to obtain sensitivity maps that are then perceived to indicate causal relationships, a methodology that has received some criticism [R1]. More directly, there has been some effort in addressing the issues for the forecasting tasks [R2].

I find that many of the similar problems listed in these papers apply here: specifically that the perturbed samples may no longer lie on the actual data manifold of the real-world data. Simply ignoring the state of an agent to be removed tampers with the validity of the rest of the agents behaviors in the scene as well. This is perhaps the most egregious in the `RemoveNoncausalEqual` perturbation where an equal number of randomly selected non-causal agents are removed. What guarantees that the scene is still valid for the model to make a prediction?

Stated differently, if the perturbation results in a different scene configuration that a model has seen before, even a perfect model might change its predictions but that would not constitute a lack of robustness. For instance, multiple valid predictions are possible for a given sequence. Why is Fig. 1 considered non-robust when there might be other examples of a left turn that seems plausible?

This in theory would not be a problem with the way causality has been mathematically defined, but I believe the problem comes from the way the definition of a causal agent is operationalized, as I describe later in point 3.

#### 2\. Accounting for the possibility of multiple valid futures:

The larger body of work of pedestrian forecasting deals with the possibility of multiple valid futures for an observations [R3, R2]. (Also see [R3, 8.4.2] for a discussion on robustness and a link to "Lasota and Shah 2017" in that paper which might be useful.) Here the notion of causality does not incorporate this possibility. The models have been trained on a large number of examples, where an agent behavior that causes a change in one example may not cause a change in another. I believe it is possible that this is what explains the large variation under non-causal perturbations as well. [R2] seems to do this in the counterfactual sense as well, but in terms of observed factors that change the 'uncertainty' over possible futures. Here the instructions do somewhat account for this by stating that a casual agent is `one whose presence would modify or influence human driver behavior in any way`.

To be fair to the authors I cannot see a directly easy way to fix this methodologically. So at the very least making this discussion explicit for future researchers would help minimise the misinformation in this space coming in from XAI methods. The linked papers might help in this regard.

#### 3\. Operationalization of causal agents and change in causality over the video:

The paper defines non-causal agents as:
```
we define a non-causal agent as an agent whose deletion does not cause the ground truth trajectory of a given target agent to change.
```

Is a causal agent then the inverse or contrapositive of this statement? i.e. is an agent causal when their presence causes a trajectory to change, or if their presence *may or may not* cause a trajectory to change?

The annotator instructions indicate it's closer to *may or may not*:

```
If the behavior of a human driver would be modified because of a potential action that an agent is likely to take, then that agent should be causal.
```

Driving as an activity inherently involves forecsating. From this definition, any potential agent that can influence one's own future is causal.

In Fig. 6. if the agent on the left (marked as not causal) rushes ahead, say under a red light, they would switch to being causal right?

But that's not how the annotation seems to be setup. The same agent can go from being causal to non-causal in principle. But I don't believe that is the case in the annotations? In the annotation statistics the authors say that "cyclists are relatively more
likely to be causal agents than pedestrians or vehicles" which seems odd as an overall statement seeing as cars around the driver ought to always be causal. If this is simply computed as `(# of agents marked as causal / # of agents in data) for each type`, this is a misleading message.

#### 4. Annotation Methodology:

#### 4.1. Annotator Agreement ility and `true' causality of agents

Since improving the reliability of models is one goal of this work, it would help to look at an inter-annotator agreement/reliability metric to see how much the annoatators agreed on an agent being causal, to understand the nature of the problem better.

The paper assumes there is an underlying truth in whether an agent is causal or not, while simultaneously acknowledging the subjectivity in the perception of the causality:

``` We emphasize that false positives (identifying an agent as causal when it is truly non-causal) are acceptable to a certain extent, but we should avoid false negatives (failing to identify a truly causal agent). ```

This subjectivity is reflected in that 24% of the labels are selected by a single annotator. The method leans towards treating false positives as accceptable, but there is no way of knowing whether these 24% would qualify as false positives or not. Traditionally in such subbjective annotations (e.g. perceived emotions) the agreement indicates wheter there is consensus in what would change a trajectory. Such a metric would provide more insight into the nature of the problem.

#### 4.2. Whose perception is being modeled?

When setting up annotation collections of this nature, the perception of annotator is asked. Here the instructions ask the annotator to simulate a hypothetical driver (Appendix A). Were all annotators drivers? Why not just ask for whether they themselves would be influenced by an agent?


[R1] Exploratory not explanatory: Counterfactual analysis of saliency maps for deep reinforcement learning - Atrey et al.
[R2] Why Did This Model Forecast This Future? Closed-Form Temporal Saliency Towards Causal Explanations of Probabilistic Forecasts - Raman et al.
[R3] Human motion trajectory prediction: A survey - Rudenko et al.

**Summary Of The Paper:**

The paper presents a new benchmark for evaluating the robustness of models for motion forecasting of autonomous vehicles. The authors collect human-annotated labels of causal agents on the Waymo Open Motion Dataset, and perform various perturbations of the data using these labels. This data is used to evaluate the robustness of state-of-the-art forecasting methods. Towards this end, the authors also propose two metrics to quantify robustness. The takeaway messages are that all models exhibit a large shift in performance under even non-causal perturbations, and that increasing the training dataset size and targeted augmentation can improve model robustness.



**Summary Of The Review:**

The paper is largely well-written and the analysis is extensive. The problems I find lie in dealing with the nature of the problem in setting up the human annotations. To be fair to the authors, if I find I cannot suggest any easy way of fixing the problems, I have not penalized my rating, and instead suggested discussing the problems as a means of increasing awareness for future researchers which is still valuable.

---

> ### Author Response · Authors · 2022-11-11
> **Response**
>
> **1. On causality and perturbation**. We appreciate the nuanced discussion and comments left by Reviewer UcfZ and we agree with many of the points raised with respect to causality and perturbations. In particular we want to acknowledge the concern that the perturbed samples may no longer lie on the actual data manifold of the real-world data.  In practice, this problem of leaving the data manifold seems less acute for this specific problem setup, which is a strength of this project and benchmark. Trajectory prediction, in the way it's formulated here (boxes in, trajectories out), is unique in that it's much easier to roughly stay in distribution removing non-causal agents. This would not be true if the models took raw sensor data.
>
> Some additional thoughts:
>
> * There are some very clear cut cases where we delete non-causal agents and observe an undesirable change in model performance. For example, in manual triage, we found examples (one of such examples is provided in Figure 11 in Appendix) where we would delete parked cars outside of the road barriers and see the model weirdly slowing down the prediction. We don’t want our predictive models used for autonomous driving to use the presence of other agents in the scene to determine driveable road regions. Moreover, we can find similar real examples to such perturbed scenarios, i.e., such perturbed data is not off the manifold of real-world data, and the slow-down prediction under the perturbation cannot be explained by multiple valid future trajectories since in the similar real-world data, we did not see more slow-down trajectories.
> * Even if the perturbed data is off the manifold of real-world data, we still want to evaluate how robust our predictive models are towards such perturbations considering the safety-critical applications. For instance, the upstream perception system might miss detection of a particular agent during inference time, and it’s good to know in advance how off the predictive model might be given such upstream errors so that we can mitigate with downstream modules to assure the safety of the overall autonomous driving system. It's also possible that non-causal perturbations can be used as a proxy for certain types of more realistic distribution shifts. Ultimately, we want to be able to test whether techniques that make models robust to non-causal perturbations help with other types of robustness.
> * We also answer the reviewer’s question regarding why we claim Figure 1 as non-robust instead of a possible future trajectory among multiple trajectories. We claim it’s not robust because ***it fails to capture the right-turn mode*** which should be included as one of the multiple valid future trajectories.
>
> **2. Accounting for the possibility of multiple valid futures**.  We specifically introduce the IoU metric in Section 4.2 to account for the possibility of multiple valid futures. The metric captures change from the original model prediction to the perturbed model prediction without using the ground truth trajectory as a reference.
>
> **3. Operationalization of causal agents and change in causality over the video**.  We annotate scenes (not timesteps) so if the agent is ever causal at any time step, it is labeled as causal for the scene. This means it is possible for causal agents to be non-causal at certain intervals in time. As defined, causal agents are those whose presence would modify or influence human driver behavior in any way (or at any point in time). We encourage labelers to be overly conservative when labeling agents as causal and to err on the side of inclusion if they are unsure. Our goal is to make it more likely that the agents we delete (the non-causal agents) will not influence the ground truth trajectory, so that we will get a relatively conservative evaluation of the robustness of the models.
>
> **4. Annotation Methodology**.
>
> **4.1 Labeler agreement**. The labeler agreement statistics are included in Appendix B. Can the reviewer clarify if there is a different type of agreement metric they would like to see? It would be interesting to look at how many false positives are present in each of the bins for Figure 7, but even the definition of what is a false positive is subjective. We use a binary label out of necessity in order to carry out the experiments, but we agree causality is inherently subjective and ambiguous and should lie on a spectrum.
>
> **4.2 Whose perception is being modeled?** One reason to ask the annotators to simulate a hypothetical human driver is that the ground truth trajectory was generated by a human driver. We want to know if deleting an agent would influence this trajectory. We did not ask if all annotators were themselves drivers, but this is a good point to add to the discussion.

---

> > ### Comment · Reviewer_UcfZ · 2022-11-18
> > **Acknowledging response**
> >
> > Hi authors,
> >
> > Just wanted to quickly acknowledge your responses. I saw them yesterday, but it's been a very short response period and I might not have time to write out a proper response before the deadline. Nevertheless, as a fellow author with zero reviewer responses, I didn't want to subject you to the same feelings of uncertainty.
> >
> > I'll leave a response if I can later today, and irrespective, promise to account for your response and participate in discussions with the other reviewers and the AC. Apologies, I've done the most justice I can with my reviews and hope you found something useful, but it's been a very hectic reviewing / rebutting period.
> >
> > Best,
> >
> > Reviewer UcfZ

---

### Official Review · Reviewer_3msf · 2022-10-25

**Confidence:** 3
**Correctness:** 3
**Technical Novelty And Significance:** 2
**Empirical Novelty And Significance:** 2
**Recommendation:** 5

**Clarity, Quality, Novelty And Reproducibility:**

 - The paper’s organizational structure can be further improved and there are points that require further clarification. Please see the above for detailed questions.
 -The perturbation is novel in terms of removing agents in the scene.


**Strength And Weaknesses:**

Strength:
 - This paper proposes a novel perturbation which is deleting agents in the scene to emphasize the causal relationship between agents.
The experiments are conducted in realistic datasets.

Weakness
 - The agents in the scene would affect the autonomous vehicle at different timesteps. In this paper, the causal agents are detected for the whole scene and are masked out during the whole scenario in the perturbed data. Is it more reasonable to label the timestep that each agent takes effects on the AV and mask accordingly?
 - Considering that this is a benchmark paper, it would be interesting to replicate the experiments in section 4.3 on other methods, such as SceneTransformer and WayTransformer.

There are several points that need further clarification.
 - For a better presentation of the two proposed metrics mentioned in the intro, It would be better to describe the IoU based metric in section 3.4 instead of in the result section.
 - It would be better to describe the definition of minADE instead of referring to the original paper, considering that minADE is quite important in evaluating the methods.
 - In section 4.2, the paper mentions that “but a long tail of outlier examples experience a large change ( > 1m).” What does “(?1m)” mean? Is the statement related to Figure 4?
 - In section 4.2, the paper mentions that “the model sees a large portion of examples where minADE improves”. What does the improvement mean? Which metric is used for measuring the improvements?
 - For figure 5, it would be better to put labels on the y-axis to make a clear comparison.
 - Typo: In page 1, “data augmentation effect model sensitivity.” -> “data augmentation affect model sensitivity.”
 - Typo: In page 5, “there are causal agents in the scene For example,” -> “there are causal agents in the scene. For example,”


**Summary Of The Paper:**

This paper constructs a novel benchmark for evaluating the robustness of trajectory prediction models by deleting non-causal agents in scenes. The causal agents in WOMD’s validation data set (around 42k scenes, as indicated in the paper) are labeled with human labelers. This paper further proposes two metrics to evaluate the robustness of models. To improve the model robustness, this paper considers data augmentation methods that show improvements over baselines.


**Summary Of The Review:**

There are several key clarifications that need to be addressed before I could fully evaluate the paper. I would recommend rejection for now.

---

> ### Author Response · Authors · 2022-11-11
> **Response**
>
> We thank the reviewer for their feedback and respond to their concerns below.
>
> **Labeling timesteps**. We agree with the reviewer that labeling each frame or timestep would be an improvement. Unfortunately, this would have significantly increased our labeling costs and time and was infeasible given our budget.
> Instead, we define an agent as causal if it impacts the ground truth trajectory at any timestep. Hence, the non-causal agents would be those that never impact the predicted agents at any timestep, and we only delete those non-causal agents in the non-causal perturbation. While labeling each frame would capture more non-causal agents at different timesteps, it won’t impact the validity of our existing non-causal labels.
>
> **Replicating experiments**. We will work on replicating the experiments in Section 4.3 on Wayformer and Scene Transformer. We don’t expect that the main takeaways will change when we vary the underlying model.
>
> **Clarifications**. Thank you for the detailed list of clarifications. We went through each one and updated the paper appropriately.  We think the organizational structure and clarity of the paper has improved as a result.
> We answer your questions directly here (in addition to updating the text):
>
> * \> 1m:  minADE is measured in meters and we are referring to a change in minADE.
> * An improvement in minADE means the minADE is lower. The sentence refers to measuring changes for individual scenes using minADE directly.

---

### Official Review · Reviewer_oUDJ · 2022-10-25

**Confidence:** 4
**Correctness:** 3
**Technical Novelty And Significance:** 3
**Empirical Novelty And Significance:** 3
**Recommendation:** 6

**Clarity, Quality, Novelty And Reproducibility:**

The paper is very well written and clear. The novelty is somewhat limited in light of prior works such as "Are socially-aware trajectory prediction models really socially-aware?, Saadatnejad et. al.". There are no issues wrt to reproducibility per se.

**Strength And Weaknesses:**

Strengths,
* The paper addresses an important problem and is well written.
* The proposed "causal" labels for Waymo Open Motion Dataset is an interesting and novel idea.
* The paper reports interesting findings: e.g. Figure 1, where there are very large changes in performance even when "non-causal" agents are perturbed.
* The paper includes extensive and fine grained evaluation on the robustness of motion forecasting models.
* The paper discusses the underlying causes of the lack of robustness of motion forecasting models in detail.

Weakness,
* The labelling process as described is prone to errors: as described each frame is annotated by only 1 annotator, this can introduce significant errors as the definition of "causal" in Appendix A subjective. E.g. in Figure 6, is it not clear why the while sedan on the right is causal? Moreover, the policy of "please err on the side of including" can introduce a significant number of false positives. This labelling policy would have a direct impact on the experiments in Section 4.1 "RemoveCausal".

* Important prior work is not discussed "Are socially-aware trajectory prediction models really socially-aware?, Saadatnejad et. al."  This work shows that small perturbations on the trajectories of agents can cause large changes in the predicted future trajectories. In the light of these finding, the findings of this paper are not very surprising.

* The paper does not propose any method to improve robustness: the utility of the paper would increase manifold if the proposed labels could be used to improve robustness.

**Summary Of The Paper:**

The paper focuses on the robustness of motion forecasting models and makes two main contributions: 1. additional labels for the Waymo Open Motion Dataset with causal labels, 2. evaluation of the robustness of several state of the art models using these "causal" labels. The causal labels include agents whose presence influences human drivers’ behaviour in any format.  The paper shows that state of the art models exhibit large shifts in performance under even non-causal perturbation: a 25-38% relative change in minADE.

**Summary Of The Review:**

The paper is an interesting read. However, the labelling process needs to be discussed in more detail especially the effect of false positive "causal" agents and its effects on the conclusions of the paper. Finally, prior works need to be discussed in more detail.

---

> ### Author Response · Authors · 2022-11-11
> **Response**
>
> We thank the reviewer for their feedback and respond to their concerns below.
>
> **Labeling process**.  As we state on page 4, Section 3.1, subheader “Human Annotations”, each scene is annotated by *at least 5 human labelers to mitigate possible errors*. Moreover, if at least one human labels an agent as causal, then we mark that agent as causal. The reviewer is correct that our labeling policy and instructions may lead to more false positives for causal agents. However, this was our intention because, for measuring robustness, we primarily care about the robustness of predictive models under the perturbation of deleting non-causal agents (i.e., agents NOT marked as causal). Hence, even with false positives for causal agents, we would get more conservative robustness evaluation results by minimizing the chance that we delete a truly causal agent.
>
> Overall, the focus of our paper is benchmarking robustness to non-causal perturbation, so we optimized to have less false positives in the non-causal labels. The RemoveCausal experiments are meant to be more of a sanity check that removing causal agents influences the behavior of the model.
>
> **Prior Work**. We thank the reviewer for pointing out the prior work of Saadatnejad et. al. and we updated our paper to include the reference. As to the novelty of our findings, the reviewer suggests that because models are sensitive to adversarial perturbations, it is not surprising that they are sensitive to our perturbations. However, this logic is flawed because (1) our non-causal perturbations are constructed differently, and (2) they are arguably simpler and more benign.
>
> In particular, Saadatnejad et. al. consider adversarial perturbations of existing trajectories where the perturbation is designed to lead to a collision. Such perturbations may be more likely to cause large changes in future trajectories as the model tries to avoid collisions, or because scenes with collisions are less common in the original data. In contrast, our work does not use an adversarial model or worst case analysis to generate perturbations. We explicitly try to create non-causal perturbations that intuitively (based on humans’ intuition / labels) should not affect the future trajectories of the predicted agent.
>
> **Methods to improve robustness**. In Section 4.3, we show that training with non-causal data augmentations and increasing dataset size improves model robustness to non-causal perturbations. For the data augmentations in particular, we use the causal labels to design the augmentations in order to improve robustness.

---

### Author Response · Authors · 2022-11-11
**Rebuttal Revision to Paper**

We thank all the reviewers and we have responded to each of their comments individually. We also updated uploaded a new version of our paper with the following changes (highlighted in red in the paper):

* Added literature review of adversarial robustness in trajectory prediction models, including all 4 papers mentioned by Reviewer vJxd
* Added a reference to "Are socially-aware trajectory prediction models really socially-aware?, Saadatnejad et. al. as mentioned by Reviewer oUDJ
* Went through each of the clarifications suggested by Reviewer 3msf and made the appropriate changes in the text

---

### Decision · Program_Chairs · 2023-01-20

**Decision:**

Reject

**Justification For Why Not Higher Score:**

1. The motivation is unclear
2. The experiments are weak.
3. Technical contributions are somehow limited.

**Justification For Why Not Lower Score:**

NA

**Metareview: Summary, Strengths And Weaknesses:**

This paper was reviewed by four experts in the field and received a mixed score. The main concerns are the limited novelty, unconvincing experiments, and lack of clarity. The authors did a good job of rebuttal and addressed many of the concerns. However, the reviewers (including all positive ones) still feel that more work is needed to get it to the best version. AC also agrees that this work can be much stronger with additional experiments. While this paper clearly has merit, the decision is not to recommend acceptance. The authors are encouraged to consider the reviewers' comments when revising the paper for submission elsewhere.